# Hybrid machine learning models for enhanced arrhythmia detection from ECG signals using autoencoder and convolution features

Subir Biswas[1]◉, Prabodh Kumar Sahoo[2]◉*, Brajesh Kumar[1]◉, Adyasha Rath[1]◉, Prince Jain[2]◉*, Ganpati Panda[3]◉, Haipeng Liu[4,5]◉, Xinhong Wang[6]◉*

1 Department of Computer Science and Engineering, C.V. Raman Global University, Bidya Nagar, Bhubaneswar, Odisha, India, 2 Department of Mechatronics Engineering, Parul Institute of Technology, Parul University, Vadodara, Gujarat, India, 3 Department of Electronics and Communication Engineering, C.V. Raman Global University, Bidya Nagar, Bhubaneswar, Odisha, India, 4 Centre for Intelligent Healthcare, Coventry University, Coventry, United Kingdom, 5 National Medical Research Association, Leicester, UK, 6 Department of Radiology, The Second Affiliated Hospital, Zhejiang University School of Medicine, Hangzhou, Zhejiang, China

◉ These authors contributed equally to this work.
* 2611104@zju.edu.cn (X.W.); princeece48@gmail.com (P.J.); sahooprabodhkumar@gmail.com (P.K.S.)

## Abstract

Automated arrhythmia detection from electrocardiogram (ECG) signals is crucial and important for the early treatment of cardiac disease (CD). In this investigation, eight machine-learning models have been developed to identify improved ECG arrhythmia detection using two standard datasets (MIT-BIH Arrhythmia and the ECG 5000). In the first phase, two types of feature extraction schemes (autoencoder) and (Convolution) are used to obtain relevant features from ECG samples and subsequently, eight ML models are successfully trained and tested to find various performance matrices through simulation-based experiments. Then, the TOPSIS and mRMR ranking schemes are used to rank the ML models and identify the three best-performing models recommended for real-time arrhythmia detection. In this study, it is observed that for the same number of input features, models based on autoencoder features offer enhanced performance compared to those based on convolutional features. It is generally observed that the top identified hybrid model, Autoencoder Features with Neural Network (AEFNN) on the MIT-BIH dataset, achieves an accuracy of 97.96% and on the ECG5000 dataset, the hybrid model achieves an accuracy of 99.20%. This proposed model can be utilized for the early detection of arrhythmia, particularly in large-scale healthcare screening programs, thereby aiding in timely diagnosis and intervention. In this study, two types of features are used to model development in future work. Other relevant important features can be extracted from ECG samples, and those features can be used to develop accurate models to identify Heart disease.

**Data availability statement:** In this research, we utilized two publicly available datasets: the MIT-BIH Arrhythmia Dataset and the ECG5000 Dataset. These datasets were instrumental in evaluating the performance of our proposed methods. - The MIT-BIH Arrhythmia Dataset can be accessed at https://www.physionet.org/content/mitdb/1.0.0/. - The ECG5000 Dataset is publicly available at https://www.timeseriesclassification.com/description.php?Dataset=ECG5000. To promote reproducibility and transparency, we have made all Python code, scripts, and additional materials used in this study publicly available. These resources can be accessed through the following GitHub repository: https://github.com/subirbiswas192001/-Arrhythmia-Detection/blob/main/README.md.

**Funding:** This work was supported by the Zhejiang Provincial Natural Science Foundation (Grant No. QN25H180024 to XW). The funders had no role in study design, data collection and analysis, decision to publish, or preparation of the manuscript.

**Competing interests:** The authors have declared that no competing interests exist.

## Introduction

Cardiac disease (CD) continues to be one of the leading causes of mortality worldwide. It is reported that one-third of total deaths in the world are due to CD. Early and proper detection of arrhythmia in ECG signals is important for effective diagnosis and treatment [1]. The various types of input signals of the patients used for the detection of CD are echocardiography, computed tomography, ECG, cardiac magnetic resonance imaging, and blood tests [2,3]. Prevailing methods of ECG analysis are based on manual interpretation by medical professionals or simple algorithmic approaches which are time-consuming and at times erroneous. In the recent past, Machine Learning (ML) and Deep Learning (DL) techniques have played important roles in the automatic and efficient detection of arrhythmias. The ML and DL Models after being trained with larger ECG datasets have the potential to identify inherent complex patterns and relationships. However, research works in the field of arrhythmia detection are still being carried out to develop and complement robust, more accurate, reliable, and less complexity-based detection models [4]. To assess the current trend of research in this field an exhaustive literature survey has been carried out and the findings are presented in the next section.

This paper presents a comparative study of two advanced deep learning techniques, namely, autoencoders and convolutional networks, for feature extraction, along with traditional machine learning methods for the early detection of ECG arrhythmia. The autoencoder was trained using the MIT-BIH dataset. The trained autoencoder was saved and then imported, with the model truncated at the bottleneck. Next, the trained encoder was used to reduce the dimensionality of the ECG data and extract essential features, which were subsequently classified using Neural Ntwork (NN) [5], Support Vector Machine (SVM) [6], Decision Tree (DT) [7], and K-Nearest Neighbors (KNN) [8], Random Forests (RF) [9] classifiers. Similarly convolutional networks were used for feature extraction employing two 1D Conv layers [10] and two 1D MaxPooling layers [11] to capture spatial features (aiming to extract the same number of features) and These extracted features were then used with NN, SVM, RF, DT, and KNN classifiers. Performance was evaluated using various metrics such as accuracy, F1-score, and AUC. To determine the superior model, the Technique for Order of Preference by Similarity to Ideal Solution (TOPSIS) [12] and Maximum Relevance Minimum Redundancy (mRMR) [13] were employed.

## Related work

A comprehensive review of the literature related to arrhythmia detection using ML and DL models is presented in this section. Shin et al. [14], have proposed an improved anomaly detection method in ECG signals using an enhanced version of AnoGAN. In another work [15], an ANNet and a lightweight neural network have been employed for ECG anomaly detection using imbalanced datasets through SMOTE augmentation. In an interesting paper [16], the authors have presented a CNN model to detect twelve types of arrhythmia from the ECG dataset. The proposed model has yielded 86% accuracy of classification. Xiu et al. [17], have presented a novel privacy-preserved model for the classification of ECG signals and have achieved an

accuracy of 93.02% using Graph Nural network (GNN). In [18], The authors explore the application of deep learning techniques, including deep neural networks and convolutional neural networks, for medical anomaly detection using diverse data sources like MEG, MRI, phonocardiographic sensing, and wearable sensors. It also highlights limitations such as poor generalization to new conditions, lack of interpretability, data scarcity, and insufficient uncertainty estimation. In another work [19], anomaly detection in ECG samples by the CNN has been carried out. The paper proposes cost-sensitive learning and employs imbalanced ECG data as input. For efficient prediction of imminent malignant ventricular arrhythmias from ECG data the authors [20], have suggested Decision Tree (DT), Naive Bayes (NB), and Support Vector Machine (SVM) based intelligence models. It is reported that eight ECG feature-based decision tree classifiers demonstrated enhanced performance in terms of sensitivity and execution time. For the detection of anomalies in ECG signals the author [21], has proposed a novel model consisting of embodying layers as well as a transformer encoder. The proposed model has been tested and validated using two datasets. It is reported that for the ECG5000 dataset, the accuracy obtained is 99%. whereas for the MIBIH ECG dataset, it is 95%.Qin et al. [22], recently have suggested Generative Adversarial Networks (GANs) models for efficient detection of the anomaly of ECG signals. Experimental results obtained from the proposed models exhibit an accuracy of 95%. In [23], a hybrid model has been proposed by employing a variational encoder and local similarity score. It is reported that the suggested model has achieved an AUC of 98.79%. For the detection of abnormality of ECG signals the authors [24], have developed two neural network models such as ConvLSTM2D-liquid time-constant network (CLTC) and ConvLSTM2D-closed-form continuous-time neural network (CCfC). The performance of those models has been evaluated with standard ECG datasets and It is observed that both models have demonstrated satisfactory detection performance. Huang et al. [24], have developed a hybrid CNN-LSTM network employing fractional wavelet transforms and principal component analysis for ECG abnormality detection.

A study [25], focuses on classifying ECG and electroencephalogram (EEG) signals using decision trees, specifically random forest and naïve Bayes algorithms. The analysis targets heartbeats and brain activity related to arrhythmia and seizures, utilizing the MIT-BIH arrhythmia and ECG database. The paper [26], introduces a real-time system for detecting ECG anomalies using an artificial neural network (ANN) with up to 99.3% accuracy. Implemented on a Raspberry Pi B+, the system collects and processes ECG data, streaming it in real time with Apache Kafka and MQTT. It alerts healthcare providers when abnormalities are detected and is evaluated on PTB and MIT-BIH datasets. This study [27], explores ways to address class imbalance in medical datasets, particularly for bio-signals like ECG, where standard data augmentation methods often fall short. By experimenting with different loss functions, data amounts, and grouping methods using the Inception-V3 model, the study proposes focal loss as an effective solution, achieving a high F1 score of 0.96.

## Problem formulation

### Research gap

Despite significant advancements in the detection of heart disease using ECG data, there remains a need for more comprehensive evaluations of various machine learning models on diverse datasets. Previous studies have often focused on a limited set of algorithms or datasets, leading to a lack of generalized understanding about which models perform best under different conditions. Furthermore, the integration of novel ranking methods such as TOPSIS and mRMR in the context of ECG data analysis has not been thoroughly explored.

### Motivation of research

The motivation for this research arises from the critical need to enhance the accuracy and reliability of heart disease detection using ECG data. With CD being one of the leading causes of mortality worldwide, improving diagnostic methods can significantly impact patient outcomes. By systematically comparing a wide range of machine learning techniques and employing advanced ranking methods, this study aims to identify the most effective models, thus contributing to the development of more precise and dependable diagnostic tools.

## Objectives of research

The main objectives of this research are:

- A comprehensive analysis contrasting two distinct methods of feature extraction, highlighting their significant differences and advancements
- To evaluate and compare the performance of various machine learning models for heart disease detection using ECG data
- To implement and analyze the efficacy of different ranking methods, specifically TOPSIS and mRMR, in assessing model performance.
- To identify the top-performing models across multiple datasets, providing a comprehensive understanding of their strengths and limitations.

## Contributions

This research makes several key contributions:

- Design an autoencoder model that can extract the same impact full feature as Convolution
- Identification of the best-performing models for each dataset, which can guide future research and practical implementations in medical diagnostics.
- The application of TOPSIS [12], and mRMR [13], ranking methods in the context of heart disease detection, demonstrating their utility in model evaluation.
- Perform McNemar's test to statistically validate the performance differences between machine learning models.
- Enhanced understanding of the relationship between model architecture and diagnostic accuracy, contributes to the optimization of machine learning approaches for ECG analysis.

## Methods and dataset

In this paper, various machine learning algorithms and two feature extraction methods have been applied, resulting in a total of 16 different combinations. This makes it challenging to identify the best model among them. Therefore, two types of ranking methods are employed. The experiments are conducted using two standard datasets. Below is a brief introduction to the methods and datasets.

For this study, a publicly available standard dataset, the MIT-BIH Arrhythmia Dataset [28], was used. This is a labeled dataset comprising a total of 14,550 samples with 187 features, where the last column represents the label. Out of the total samples, 10,505 (size: 10,505 × 188) correspond to heart disease-affected individuals, and 4,045 (size: 4,045 × 188) represent healthy individuals. Additionally, another standard dataset named the ECG5000 dataset [29] was utilized to evaluate the model's robustness. The original "ECG5000" dataset is a 20-hour ECG recording obtained from PhysioNet, specifically the "chf07" record in the BIDMC Congestive Heart Failure Database (chfdb). Table 1 provides detailed information about the datasets used in this study, offering a comprehensive overview to enhance understanding and facilitate deeper insights.

To train the model, 70% of the data was allocated for training and 30% for testing. Table 2 illustrates the distribution of the training and testing data.

**Table 1**. Details of two standard datasets on ECG signals.

| Dataset Name | normal | abnormal |
| --- | --- | --- |
| MIT-BIH Arrhythmia | 10505 | 4045 |
| ECG5000 | 3497 | 1500 |

**Table 2**. Comprehensive overview of train and test data utilized in the study.

| Dataset Name | Training | Test |
|---|---|---|
| MIT-BIH Arrhythmia | 11,640 | 4365 |
| ECG5000 | 3997 | 1500 |

## Feature extraction methods

Two different types of feature extraction methods have been demonstrated in this study A brief introduction of that method is discussed below.

**Autoencoder.**  Inspired by our brains' ability to compress information, autoencoders [21] are a special type of neural network adept at uncovering hidden patterns within 1D data, like time series or sensor readings. Imagine feeding an autoencoder a long sequence of numbers representing temperature readings throughout the day. It would then learn to compress this sequence, capturing its key trends and variations. By attempting to reconstruct the original sequence from this compressed form (latent space), the autoencoder hones its ability to identify and encode the most crucial aspects of the data. This process unlocks a treasure trove of applications specifically suited for 1D data.

Autoencoders can be used for anomaly detection, readily flagging unusual temperature spikes or dips that deviate from the learned patterns. They can also be used for data compression, efficiently storing long sequences in a more compact form. Furthermore, autoencoders even have the potential for data denoising, helping to remove noise from sensor readings and reconstruct cleaner versions of the original data. As research progresses, autoencoders promise to become even more powerful tools for analyzing and manipulating 1D data across various scientific and engineering fields [30].

**Convoluction operation.**  In this research, one-dimensional convolution was used. It is used in various signal processing and sequence modeling tasks. Unlike its two-dimensional counterpart, which operates on images, 1D convolution processes one-dimensional sequences such as time series data, audio signals, or text.

In 1D convolution [10], a filter (also called a kernel) slides along the input sequence, computing a dot product between the filter weights and the values in the input sequence at each position. This operation captures local patterns or features within the sequence. By applying multiple filters and stacking convolutional layers, the network can learn hierarchical representations of the input data, extracting increasingly complex features as the depth of the network increases. One of the key advantages of 1D convolution is its ability to capture temporal dependencies and patterns within sequential data. For example, in natural language processing tasks, 1D convolutional neural networks (CNN) can effectively learn patterns of words or characters in a sentence, capturing features like n-grams or syntactic structures.

Additionally, 1D convolution can be used in combination with other layers such as pooling and fully connected layers to build deep neural network architectures for tasks such as sentiment analysis, speech recognition, and even genomic sequence analysis.

## Machine learning models

**Support Vector Machines SVMs.**  SVMs [6], are powerful supervised learning algorithms that shine in classification tasks. Imagine data points representing different categories, like emails classified as spam or not spam. SVMs draw a dividing line (or hyperplane in higher dimensions) that best separates these categories. The key is maximizing the margin, the space between the hyperplane and the closest data points from each class (support vectors). This approach makes SVMs robust and effective, especially for high-dimensional data. SVMs can even handle complex, non-linear data using a technique called the kernel trick. Overall, SVMs [6,31], are a versatile tool for classification problems across various fields, from finance and handwriting recognition to bioinformatics and potentially diagnosing diseases based on gene expression analysis. In this simulation-based observation, with different kernels and calculated the performance for better results.

**K-Nearest Neighbors.** K-Nearest Neighbors (KNN) [8,32], is a simple yet powerful non-parametric machine learning algorithm used for both classification and regression tasks. It works by classifying a new data point based on the majority vote of its K nearest neighbors in the training data. Here, "K" is a user-defined parameter that determines the number of neighbors to consider. Distance metrics like Euclidean distance are used to measure the closeness between data points. Equation (Distance):

$$d(\mathbf{x}, \mathbf{y}) = \sqrt{(x_1 - y_1)^2 + (x_2 - y_2)^2 + \cdots + (x_n - y_n)^2} \tag{1}$$

(where x and y are data points with n features) This equation calculates the Euclidean distance between two data points ($\mathbf{x}$ and $\mathbf{y}$) with $n$ features ($x_1$, $x_2$, ..., $x_n$ and $y_1$, $y_2$, ..., $y_n$).

KNN is easy to understand and implement, making it a popular choice for various applications. However, it can be computationally expensive for large datasets and sensitive to the choice of K and the distance metric used.

**Random Forest.** Random Forests (RF) [9], are powerful ensemble learning algorithms known for their accuracy and robustness in classification and regression tasks. They operate by building a multitude of decision trees on random subsets of features and data points. Each tree predicts a class or value, and the final prediction is the majority vote (classification) or average (regression) of these individual tree predictions. This approach reduces variance and prevents overfitting, a common problem in machine learning [33].

Equation (Gini Index - Classification):

$$Gini(t) = 1 - \sum (p_i)^2 \tag{2}$$

(where t is a node in the tree and pi is the proportion of data points in class i at node t)

This equation, the Gini impurity measure, is commonly used in RF for classification. It calculates the probability of a data point at a node being misclassified if randomly labeled based on the class distribution at that node. The decision tree aims to split data points to minimize the Gini index, leading to purer leaf nodes.

**Decision trees.** Decision Trees (DTs) [7], are a fundamental supervised learning algorithm excelling at classification tasks. They work by building a tree-like structure where internal nodes represent features and branches represent possible outcomes. Each internal node applies a splitting rule based on a specific feature, separating data points into more homogeneous subsets. This process continues until reaching "leaf nodes" containing the final classifications. Equation (Information Gain):

$$InformationGain(A, S) = Entropy(S) - \sum \frac{|A(v)|}{|S|} * Entropy(A(v)) \tag{3}$$

(where A is a feature, S is the dataset, and v is a value of A)

This equation calculates the information gain achieved by splitting the dataset (S) based on a feature (A). Entropy (S) measures the overall uncertainty in the dataset before splitting. The lower the information gain, the less informative the split based on that feature.

DTs are interpretable due to their tree structure, allowing for an easy understanding of decision-making processes. They can handle both categorical and continuous features without complex transformations.

**Artificial neural networks.** Neural Networks (NN) [5,34], are powerful machine-learning models inspired by the structure and function of the human brain. They consist of interconnected layers of artificial neurons, where each neuron performs a simple mathematical operation on its inputs. These layers progressively transform the data, allowing NN to learn complex patterns from large datasets.

One key equation in NNs is the activation function, which introduces non-linearity into the network. A common example is the sigmoid function [35],:

$$f(x) = \frac{1}{1 + e^{-x}} \tag{4}$$

Here, f(x) represents the output of the neuron, and x is the weighted sum of its inputs. The sigmoid function transforms the input value (x) into an output between 0 and 1.

NNs excel at tasks like image recognition, natural language processing, and time series forecasting. However, they can be computationally expensive to train and require careful design and tuning of their architecture.

**Ranking methods used for ranking the performances**

Identifying the best-performing model among a total of 16 method combinations is challenging. To address this issue, two different ranking methods were used. Below is a brief introduction to these techniques.

- **Technique for Order of Preference by Similarity to Ideal Solution (TOPSIS)**

  Çelikbilek et al. [12] Let $m$ be the number of alternatives and $n$ be the number of criteria. For each alternative $i$, let $x_{ij}$ denote the performance score of alternative $i$ on criterion $j$. Additionally, let $w_j$ represent the weight assigned to criterion $j$, where $\sum_{j=1}^{n} w_j = 1$. The normalized decision matrix is denoted by $r_{ij}$, calculated as

$$r_{ij} = \frac{x_{ij}}{\sqrt{\sum_{i=1}^{m} x_{ij}^2}}. \tag{5}$$

  The ideal solution $A^+$ and the worst solution $A^-$ are determined as follows:

$$A^+ = \left(\max(r_{ij})\right)_{n\times1} \tag{6}$$
$$A^- = \left(\min(r_{ij})\right)_{n\times1} \tag{7}$$

  The Euclidean distance between each alternative $i$ and the ideal solution $A^+$ is calculated as

$$D_i^+ = \sqrt{\sum_{j=1}^{n}(r_{ij} - A_j^+)^2} \tag{8}$$

  Similarly, the distance from the worst solution $A^-$ is calculated as

$$D_i^- = \sqrt{\sum_{j=1}^{n}(r_{ij} - A_j^-)^2} \tag{9}$$

  Finally, the relative closeness to the ideal solution for each alternative $i$, denoted by $C_i$, is computed as:

$$C_i = \frac{D_i^-}{D_i^+ + D_i^-} \tag{10}$$

  The alternative with the highest value of $C_i$ represents the best choice according to the TOPSIS method.
- **Maximum Relevance Minimum Redundancy (mRMR)**

  The mRMR method, originally proposed by Ding and Peng [13], is a technique widely used for feature selection in machine learning. It aims to identify a subset of features that are both highly relevant to the target variable and minimally redundant with each other. Relevance is typically measured using mutual information between features and the target, while redundancy is measured among the features themselves. This approach helps ensure that selected features contribute unique and informative signals to the prediction task.

In this study, we adapt the mRMR method to the context of *model ranking* based on multiple performance metrics (AUC, F1-score and Accuracy). In our adaptation, we treat each machine learning model as a "sample" and each performance metric as a "feature". The objective is to identify a subset of evaluation metrics that are highly informative for distinguishing well-performing models while minimizing redundancy among them. By applying mRMR in this setting, we are able to compute a reduced metric set that captures the most critical and independent aspects of performance.

This refined metric subset is then used as input to the TOPSIS method for final model ranking. The mRMR-based filtering ensures that the ranking process emphasizes diverse and non-overlapping performance characteristics, addressing potential biases caused by correlated metrics and offering a more balanced evaluation of model performance.

## Proposed approach

The working procedure for heart disease detection, as illustrated in Fig 1, begins with extracting relevant features. For feature extraction, two methods were applied: Convolution Operation Features (CF) and features from the encoder part of an autoencoder (AEF). In this research, the MIT-BIH dataset was used to train the autoencoder. After training, we did experiments using MITBIH and ECG5000 datasets for classification. For classification, we do experiments with 16 different combinations. First, features were extracted using an autoencoder and then classified using an SVM with 'rbf', 'poly', 'sigmoid', and 'linear' kernel (AEFSVMR), (AEFSVMP), (AEFSVMS), (AEFSVML). Similarly, Features feature extraction autoencoder and classified with KNN (AEFKNN). In the same vein, Feature extraction with an autoencoder and then classified with DT (AEFDT). Equally, Feature extraction with autoencoder and classified with NN (AEFNN). Also, Feature extraction with autoencoder and classified with RF (AEFRF).

Secondly, features were extracted using convolution. and classified by SVM with 'rbf', 'poly', 'sigmoid', and 'linear' kernel (CFSVMR), (CFSVMP), (CFSVMS), (CFSVML). Likewise, the feature can be extracted using Convolution and classified with RF (CFRF). In the same vein extract the feature using convolution and classify with DT (CFDT). Equally, the feature is extracted using convolution and classified with KNN (CFKNN). Lastly, experiments were conducted using a convolutional neural network (CFNN).

After finding the performance matrices like F1-Score, Accuracy, and AUC from these sixteen models. It is difficult to rank which model is the best-performing model. In this research, we employed two Ranking methods TOPSIS [12] and mRMR [13] for ranking the models based on their performance matrices.

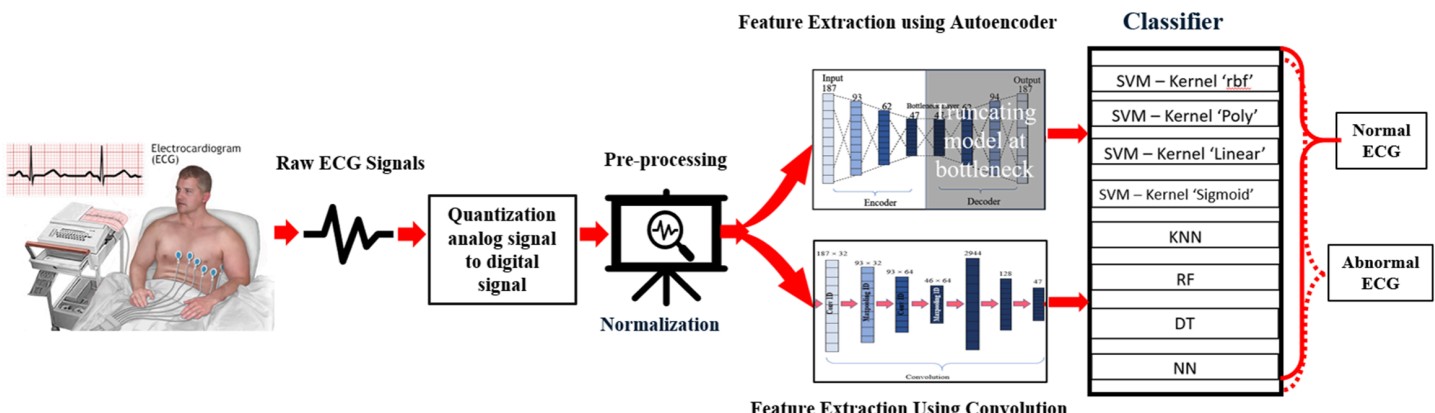

**Fig 1**. Generalised block diagram for heart disease classification by machine learning models from ECG records.

## Train Autoencoder Models for Feature Extraction (AEF)

At first, data prepossessed with a standard scale and then fit the autoencoder model. Fig 2 shows the proposed autoencoder model

**Defining autoencoder architecture used for the experiment:**

- **Input Layer:** The visible variable defines the input layer with the shape of the input data (X_train.shape[1]). X_train is assumed to be the training data. Size of (X_train.shape[1]) = 187 as input size.
- **Encoder Layers:** Several fully connected (Dense) layers followed by Batch Normalization and Leaky ReLU [36], activation functions form the encoder part of the autoencoder. Each layer gradually reduces the dimensionality of the input.
- **Bottleneck Layer:** This layer is a bottleneck or compressed representation of the input data. It has a smaller number of neurons compared to the encoder layers. The size in this layer is 47.
- **Decoder Layers:** The decoder part mirrors the encoder, with Dense layers, Batch Normalization [37], and Leaky ReLU [36] activation functions. These layers gradually reconstruct the original input shape.
- **Output Layer:** The output layer produces the reconstructed input. It uses a linear activation function.
- **Model Creation and Compilation:** autoencoder is created using the Model class, specifying the input and output layers. The autoencoder model is compiled using the Adam optimizer with a specified learning rate (0.001) and Mean Squared Error (MSE) [38] loss function.

Input data has a total of 187 features after using the autoencoder, we received 47 features, which are approximately four times less than the input size and calculated loss Fig 3 shows the training and validation loss of our autoencoder model. We calculated the loss for different Bottleneck Layer features but for feature 47 we received the minimum loss.

After 1000 epochs, our autoencoder is trained, the training loss is 0.08, and we stopped training and saved our model for future use.

**Autoencoder with machine learning model for classification.** To perform autoencoder with machine learning at first we make preprocessing with standard scale reason while training the autoencoder model with standard scale data. After pre-processing the data, we pass the normalized data into the autoencoder by the model, truncated at the bottleneck for extracting features. After getting the features, we passed the data through SVM with four different kernels, KNN,

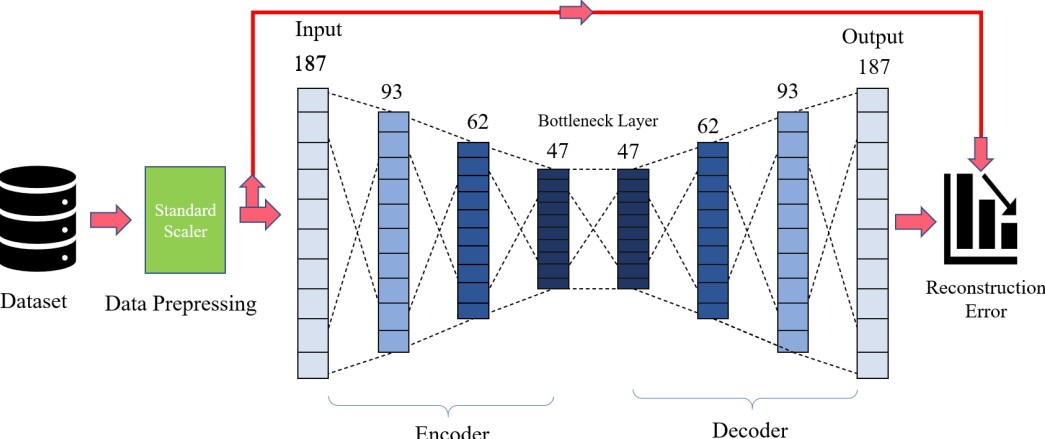

**Fig 2**. The structure of the proposed autoencoder utilized for feature extraction from ECG data.

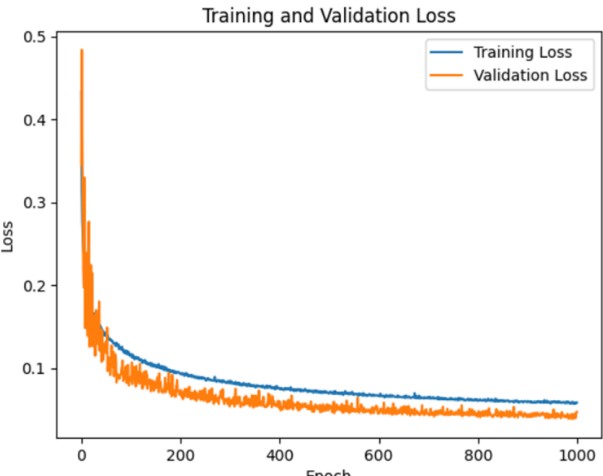

**Fig 3**. **Training and validation loss plot during training the autoencoder.**

DT, NN, and RF, then calculated the accuracy and other performance metrics. Fig 4 demonstrates the proposed work, Feature extraction by autoencoder and classification with machine learning models.

## Feature Extraction with CNN (CF)

We employed a CNN to extract features for subsequent classification using ML models. To ensure methodological consistency, we aimed to extract a comparable number of features to those obtained using an autoencoder (47 features). The CNN was specifically designed and trained to serve as a shared feature extractor across all classifiers, leveraging pre-trained convolutional layers to enhance efficiency and performance.

**Model Architecture:** The feature extraction model was implemented using the Keras API, integrated within Tensor-Flow. The architecture begins with a 1D convolutional layer [10] comprising 32 filters, a kernel size of 3, a ReLU activation function, and 'same' padding. The input shape was defined based on the dimensions of the training data. This was followed by a 1D max pooling layer configured with a pool size of 3, strides of 2, and 'same' padding to reduce the spatial

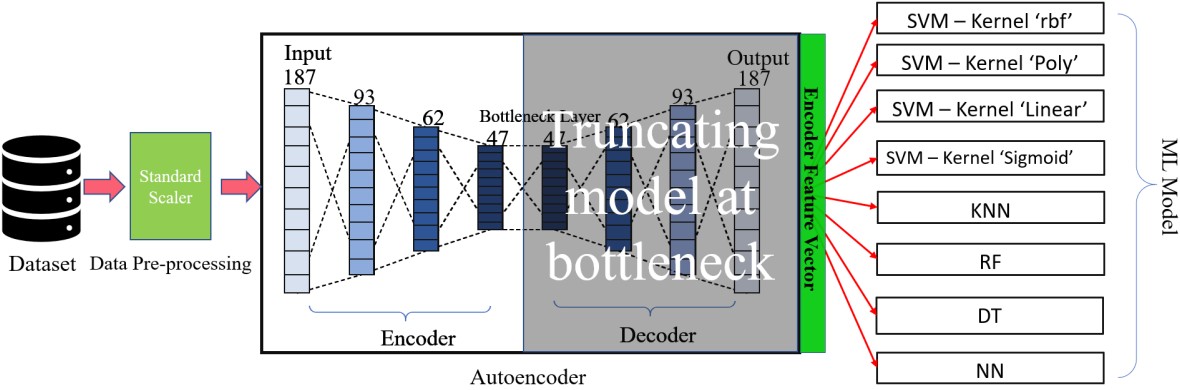

**Fig 4**. **Block diagram of encoder of trained autoencoder-based classifiers.**

dimensions. Subsequently, a second 1D convolutional layer was added with 64 filters, a kernel size of 3, a ReLU activation function, and 'same' padding. Another 1D max pooling layer [39] with the same configuration as the first was incorporated. To mitigate overfitting, a dropout layer with a rate of 0.5 was included, and the output was then flattened to generate feature vectors.

**Feature Extraction Procedure:** The CNN layers were trained once on the training dataset and subsequently reused across all classifiers, ensuring consistent feature extraction. By reusing the pre-trained convolutional layers, the model was able to efficiently extract meaningful and robust features from the input data. These features were then standardized using the 'StandardScaler' from 'scikit-learn' to ensure compatibility with ML classifiers.

The standardized features were passed to classifiers such as SVM to evaluate performance. This shared feature extraction approach not only streamlined the methodology but also leveraged the representational capacity of the pre-trained CNN, enhancing the overall classification performance.

Fig 5 illustrates the architecture of the proposed CNN model and its integration with ML classifiers.

## Simulation based experimental results

The objective of this research is to evaluate the efficacy of various classification algorithms in order to identify the most precise algorithm for predicting the likelihood of a patient developing heart disease. This study employs advanced analytical techniques to achieve its goals.

Table 3 shows the classification matrices of our proposed 16-combination model using an autoencoder with ML models and Convolution with ML models for the MITBIH dataset. The classification matrices present in this table are Accuracy, F1 Score, and AUC score. Most of the models score above 90% accuracy with autoencoder features on test data, and that indicates the robustness and reliability of the models. Table 4 shows the classification matrices of our proposed 16-combination model using autoencoder with ML models and Convolution with ML models for the ECG5000 dataset. The classification matrices present in this table are Accuracy, F1 Score, and AUC.

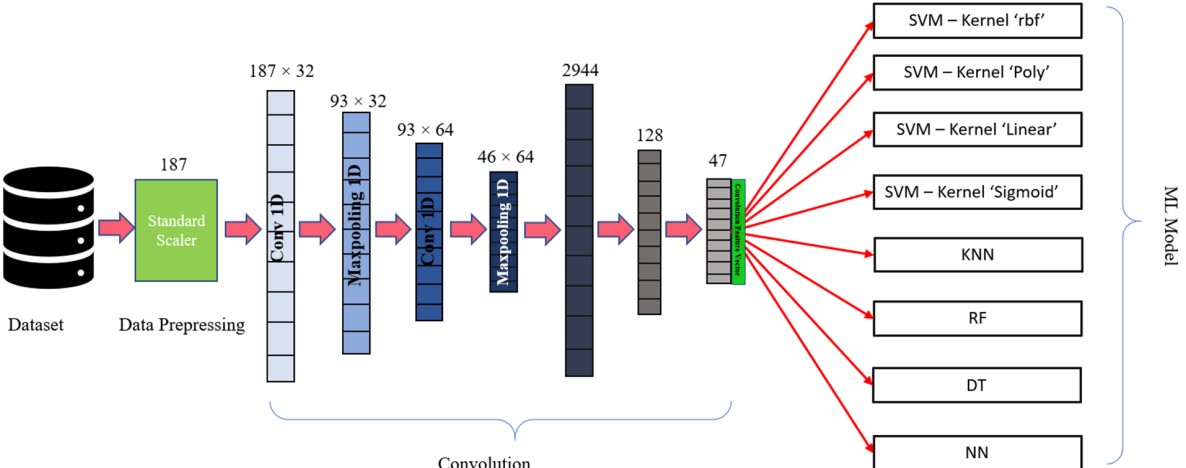

**Fig 5. Convolution with machine learning models for classification.**

**Table 3**. Comparative performance metrics of 16 ML models used two types of features for ECG records from the MIT-BIH dataset.

| SN | Method | Auc | f1-score | Accuracy |
|----|--------|-----|----------|----------|
| 1 | AEFSVMR | 90.12 | 95.00 | 92.27 |
| 2 | AEFSVMP | 90.12 | 95.00 | 92.27 |
| 3 | AEFSVMS | 65.15 | 80.00 | 71.75 |
| 4 | AEFSVML | 75.37 | 88.00 | 82.52 |
| 5 | AEFKNN | 92.67 | 96.00 | 93.72 |
| 6 | AEFDT | 83.37 | 90.00 | 86.34 |
| 7 | AEFNN | 97.57 | 99.00 | 97.96 |
| 8 | AEFRF | 91.11 | 96.00 | 93.72 |
| 9 | CFNN | 75.11 | 86.00 | 79.42 |
| 10 | CFSVMS | 68.90 | 83.00 | 75.28 |
| 11 | CFSVMR | 73.92 | 86.00 | 79.33 |
| 12 | CFSVMP | 73.92 | 86.00 | 79.33 |
| 13 | CFSVML | 73.96 | 86.00 | 79.28 |
| 14 | CFRF | 65.88 | 81.00 | 72.14 |
| 15 | CFDT | 65.92 | 81.00 | 71.16 |
| 16 | CFKNN | 69.66 | 84.00 | 72.21 |

**Table 4**. Comparative performance metrics of 16 ML models used two types of features for ECG records from the ECG5000 dataset.

| SN | Method | Auc | f1-score | Accuracy |
|----|--------|-----|----------|----------|
| 1 | AEFSVMR | 99.33 | 99.00 | 99.33 |
| 2 | AEFSVMP | 99.33 | 99.00 | 99.33 |
| 3 | AEFSVMS | 91.95 | 93.00 | 92.00 |
| 4 | AEFSVML | 98.32 | 99.00 | 98.46 |
| 5 | AEFKNN | 98.55 | 99.00 | 99.20 |
| 6 | AEFDT | 97.92 | 98.00 | 97.93 |
| 7 | AEFNN | 98.55 | 99.00 | 99.20 |
| 8 | AEFRF | 99.22 | 99.00 | 99.20 |
| 9 | CFNN | 82.33 | 88.00 | 84.39 |
| 10 | CFSVMS | 73.77 | 79.00 | 74.73 |
| 11 | CFSVMR | 81.97 | 88.00 | 84.66 |
| 12 | CFSVMP | 81.97 | 88.00 | 84.66 |
| 13 | CFSVML | 82.57 | 88.00 | 85.00 |
| 14 | CFRF | 76.71 | 81.00 | 77.77 |
| 15 | CFDT | 76.71 | 81.00 | 77.77 |
| 16 | CFKNN | 80.50 | 86.00 | 77.73 |

## Discussions

In this study, we use various method combinations of Machine learning approaches in a total of 16, so it isn't easy to rank the results manually. That's why we use 2 different types of performance ranking methods based on three parameters AUC, F1-Score, and Accuracy. Ranking methods are TOPSIS and another mRMR. After that, we performed three simulation-based experiments to analyze this result better. Fig 6 shows the simulation-based First Experiment, and simulation-based Second Experiment, and Fig 13 shows the Simulation-Based Third Experiment.

Fig 6 shows the Systematic block cum flow diagram to identify the best performing ML models for Heart disease detection. for MITBIH and ECG5000 datasets, we flow the same procedure. The process begins with selecting an appropriate dataset, followed by feature extraction using either AEF or CF methods. Next, machine learning models are applied to classify the data. To evaluate performance, three metrics are calculated for each model. Based on these metrics, models are ranked using a model ranking method, and finally, the three best-performing machine learning models are identified.

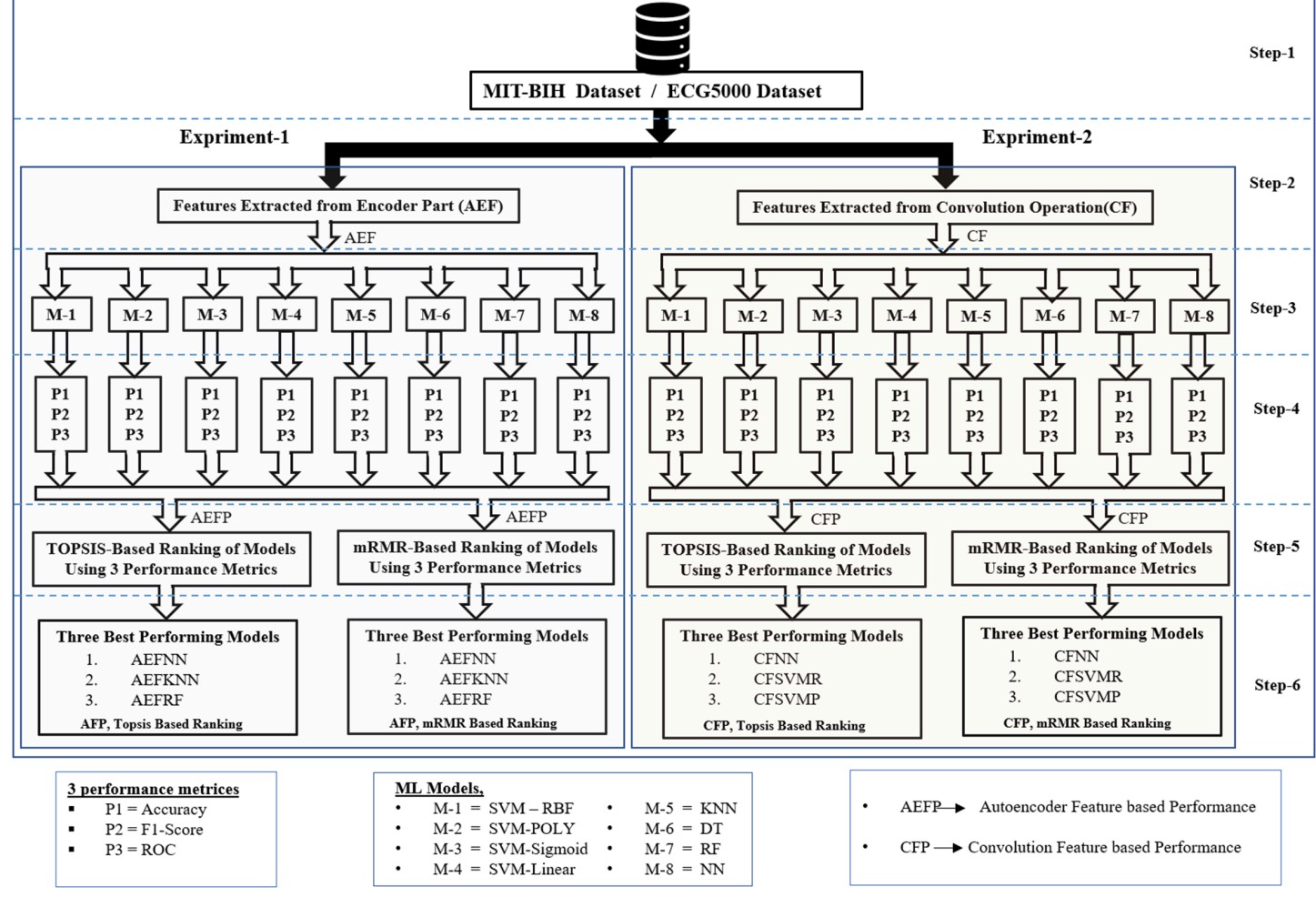

**Fig 6**. Systematic block cum flow diagram to identify best performing ML models for heart disease detection from ECG signal of patient.

## Simulation-based first experiment

During the first experiment extracted the feature with the help of an autoencoder and classified it using different ML models. Next, ranking them using Topsis and mRMR. Fig 6 showed the First Experiment steps. Feature Extraction Using autoencoder and classification using ML models Based on the Autoencoder Feature Extraction method.

Table 5 showed Our findings From the First Experiment, Ranking of Different ML Models Using Topsis and mRMR Ranking Approaches Employing a Feature set Autoencoder Obtained from ECG Signals of MITBIH Dataset. The three best-performing models in this analysis are AEFNN, AEFKNN, and AEFRF. The AEFNN model achieved the highest accuracy at 97.96%, followed by AEFKNN and AEFRF achive a similar result 93.72% accuracy.

ALL three ranking methods consistently ranked the same models as the top 1, 2, and 3. Fig 7 shows the loss plot of the best-performing model AEFNN during the training phase for Heart Disease Detection for the MITBIH Dataset. Fig 8 presents comparative ROC plots of the best and worst-performing models for heart disease detection using AEF with ML models on the MIT-BIH dataset.

**Table 5**. Comparative ranking of different ML models using Topsis and mRMR ranking approaches employing feature set autoencoder obtained from ECG signals of MITBIH dataset.

| Index | Method | AUC | F1-score | Accuracy | Rank (mRMR) | Rank (Topsis) |
|---|---|---|---|---|---|---|
| 1 | AEFNN | 97.57 | 99 | 97.96 | 1 | 1 |
| 2 | AEFKNN | 92.67 | 96 | 93.72 | 2 | 3 |
| 3 | AEFRF | 91.11 | 96 | 93.72 | 3 | 3 |
| 4 | AEFSVMR | 90.12 | 95 | 92.27 | 4.5 | 5 |
| 5 | AEFSVMP | 90.12 | 95 | 92.27 | 4.5 | 5 |
| 6 | AEFDT | 83.37 | 90 | 86.34 | 6 | 6 |
| 7 | AEFSVML | 75.37 | 88 | 82.52 | 7 | 7 |
| 8 | AEFSVMS | 65.15 | 80 | 71.75 | 8 | 8 |

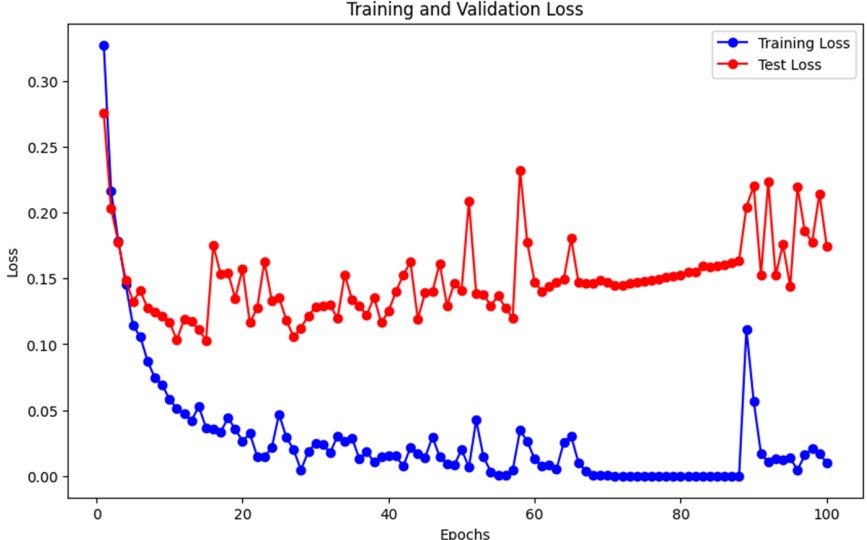

**Fig 7**. The loss plot of the AEFNN model for heart disease detection using autoencoder features on MITBIH dataset.

Table 6 presents our findings from the first experiment, which involved ranking various machine learning models using the TOPSIS and mRMR ranking approaches. The models were evaluated using a feature set obtained through an autoencoder applied to ECG signals from the ECG5000 dataset.

A total of five hybrid intelligence models achieved the highest accuracy of 99%. These models are AEFSVMR, AEFSVMP, AEFSVMRF, AEFRF, AEFKNN, and AEFNN. Among them, AEFSVMR and AEFSVMP achieved identical results in terms of accuracy, F1-score, and AUC. Similarly, AEFRF, AEFKNN, and AEFNN attained the same accuracy, although their F1-scores and AUC values varied. Notably, AEFKNN and AEFNN shared identical scores across accuracy, F1-score, and AUC. Fig 9 depicts the loss plot of the AEFNN model during the training phase for heart disease detection on the ECG5000 dataset. Fig 10 shows comparative ROC plots of the best-performing model AEFSVMR, and the worst-performing model, AEFSVMS for Heart Disease Detection using Autoencoder Features on ECG5000 Dataset.

## Simulation-based second experiment

In the second experiment, features were extracted with the help of Convolution and classified it using different ML models. Next, ranking them using Topsis and mRMR. Fig 6 shows the Feature Extraction Using Convolution and classification using ML models. The three top-performing models on the MIT-BIH dataset are CFNN, CFSVMR, and CFSVMP.

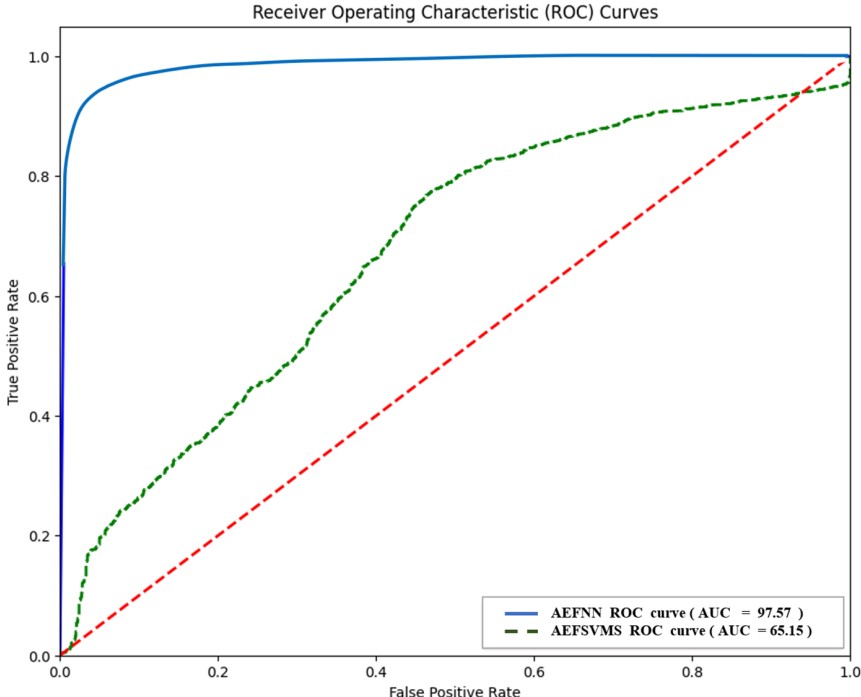

**Fig 8**. Comparative ROC plots of the best and the worst performing ML heart disease detection model for autoencoder features on MITBIH dataset.

**Table 6**. Comparative ranking of different ML models using Topsis and mRMR ranking approaches employing feature set autoencoder obtained from ECG signals of ECG5000 dataset.

| Index | Method | AUC | F1-score | Accuracy | Rank (mRMR) | Rank (Topsis) |
|---|---|---|---|---|---|---|
| 1 | AEFSVMR | 99.33 | 99 | 99.33 | 1.5 | 2 |
| 2 | AEFSVMP | 99.33 | 99 | 99.33 | 1.5 | 2 |
| 3 | AEFRF | 99.22 | 99 | 99.20 | 3 | 5 |
| 4 | AEFKNN | 98.55 | 99 | 99.20 | 4.5 | 5 |
| 5 | AEFNN | 98.55 | 99 | 99.20 | 4.5 | 5 |
| 6 | AEFSVML | 98.32 | 99 | 98.46 | 6 | 6 |
| 7 | AEFDT | 97.92 | 98 | 97.93 | 7 | 7 |
| 8 | AEFSVMS | 91.95 | 93 | 92.00 | 8 | 8 |

CFNN achieved the highest accuracy of 79.42%, followed closely by both CFSVMR and CFSVMP, each with an accuracy of 79.33%. The models CFSVMR and CFSVMP also exhibited similar performance in terms of AUC, F1-score, and accuracy. In total, four models achieved approximately 79% accuracy: CFNN, CFSVMR, CFSVMP, and CFSVML.

Table 7 showed Our findings From the second Experiment Ranking of Different ML Models Using Topsis and mRMR Ranking Approaches Employing Feature set convolution Obtained from ECG Signals of the MITBIH Dataset. Fig 11 illustrates the comparative ROC plots of the best-performing model and the worst-performing model for heart disease detection using convolutional features on the MIT-BIH dataset.

Table 8 showed our findings from the second Experiment, Ranking of Different ML Models Using Topsis and mRMR Ranking Approaches Employing Feature set Convolution Obtained from ECG Signals of ECG5000 Dataset. The best-performing models in this analysis are CFSVML, CFSVMR, CFSVMP, and CFNN, with accuracies of 85.00%, 84.66%, 84.66%, and 84.39%, respectively. The models CFSVMR and CFSVMP demonstrated similar performance in terms of

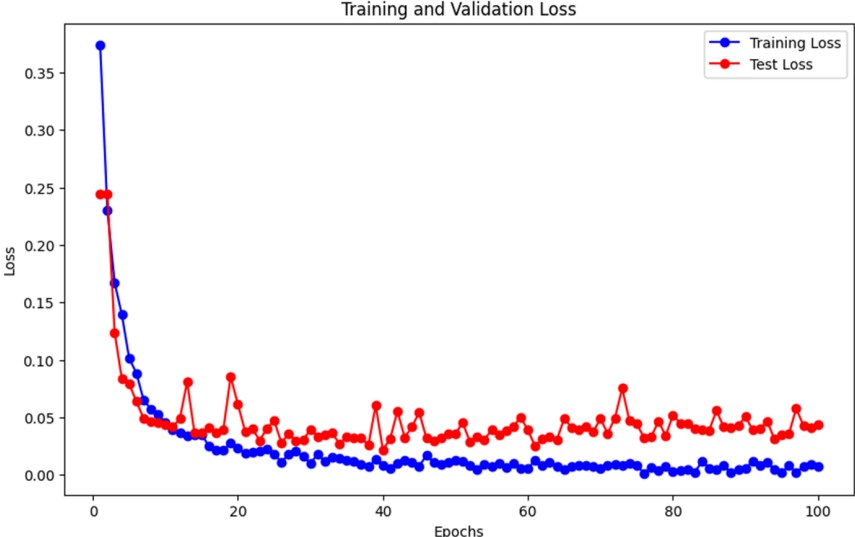

**Fig 9**. **The loss plot of the AEFNN model for heart disease detection using autoencoder features on the ECG500 dataset.**

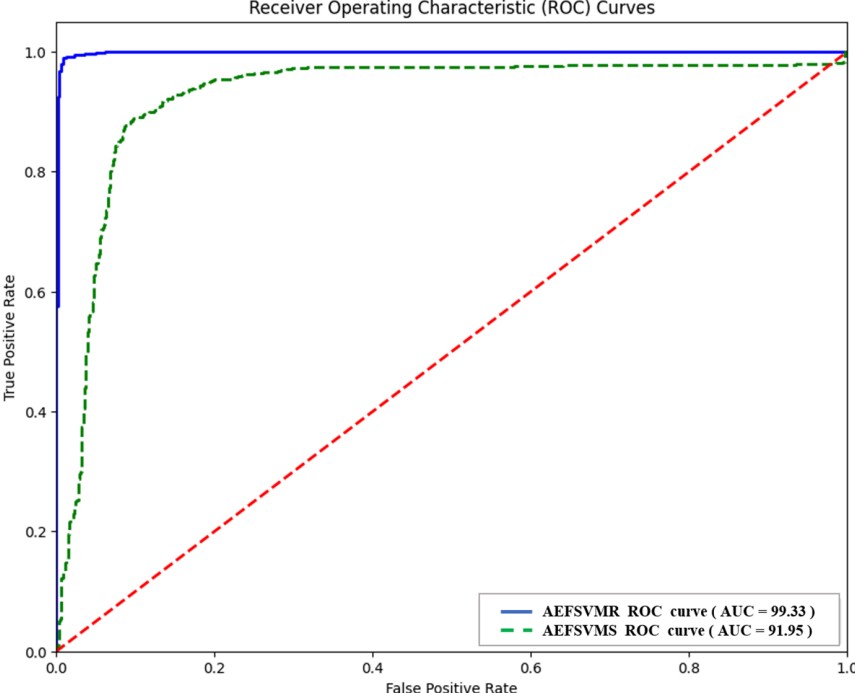

**Fig 10**. **Comparative ROC plots of the best and the worst performing ML heart disease detection models for autoencoder features on ECG5000 dataset.**

**Table 7**. Comparative ranking of different ML models using Topsis and mRMR ranking approaches employing feature set convolution obtained from ECG signals of MITBIH dataset.

| Index | Method | AUC | F1-score | Accuracy | Rank (mRMR) | Rank (Topsis) |
|---|---|---|---|---|---|---|
| 1 | CFNN | 75.11 | 86.00 | 79.42 | 1 | 1 |
| 2 | CFSVMR | 73.92 | 86.00 | 79.33 | 2.5 | 3 |
| 3 | CFSVMP | 73.92 | 86.00 | 79.33 | 2.5 | 3 |
| 4 | CFSVML | 73.96 | 86.00 | 79.28 | 4 | 4 |
| 5 | CFSVMS | 68.90 | 83.00 | 75.28 | 5 | 5 |
| 6 | CFKNN | 69.66 | 84.00 | 72.21 | 6 | 6 |
| 7 | CFDT | 65.92 | 81.00 | 72.16 | 7 | 7 |
| 8 | CFRF | 65.88 | 81.00 | 72.14 | 8 | 8 |

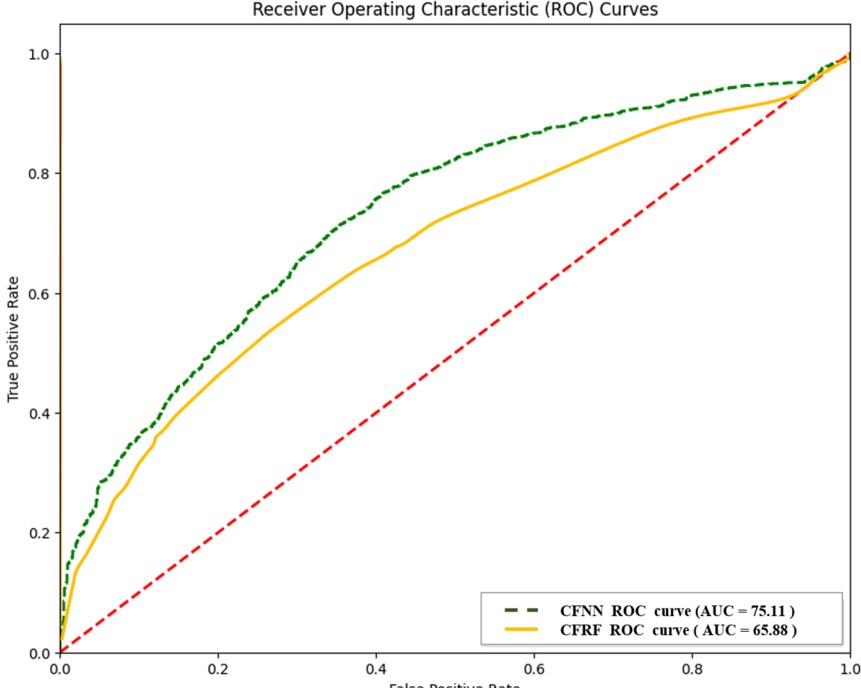

**Fig 11**. Comparative ROC plots of the best and the worst performing ML heart disease detection models for convolution features for MITBIH dataset.

AUC, F1-score, and accuracy. According to the mRMR ranking method, CFNN was ranked second, while CFSVMR and CFSVMP were ranked third. CFNN achieved an accuracy of 84.39%, an F1-score of 88, and an AUC of 82.33. In contrast, CFSVMR and CFSVMP both achieved an accuracy of 84.66%, an F1-score of 88, and an AUC of 81.97. However, the TOPSIS method assigned a ranking score of three to both CFSVMR and CFSVMP, while CFNN was ranked fourth.

Fig 12 presents the comparative ROC plots of the best and worst-performing machine learning models for heart disease detection using convolutional features on the ECG5000 dataset.

### Simulation-based third experiment

To facilitate a more rigorous comparison of the models, we standardized the dataset and employed the Topsis and mRMR methods to rank the all model's results. This allowed us to identify the highest-performing models across all methodologies. Fig 13 shows the Third Experiment procedure. Table 9 shows the comparison of the models for the MITBIH

**Table 8**. Comparative ranking of different ML models using Topsis and mRMR ranking approaches employing feature set convolution obtained from ECG signals of ECG5000 dataset.

| Index | Method | AUC | F1-score | Accuracy | Rank (mRMR) | Rank (Topsis) |
|---|---|---|---|---|---|---|
| 1 | CFSVML | 82.57 | 88 | 85.00 | 1 | 1 |
| 2 | CFSVMR | 81.97 | 88 | 84.66 | 3.5 | 3 |
| 3 | CFSVMP | 81.97 | 88 | 84.66 | 3.5 | 3 |
| 4 | CFNN | 82.33 | 88 | 84.39 | 2 | 4 |
| 5 | CFKNN | 80.50 | 86 | 77.73 | 5 | 5 |
| 6 | CFRF | 76.71 | 81 | 77.77 | 6.5 | 7 |
| 7 | CFDT | 76.71 | 81 | 77.77 | 6.5 | 7 |
| 8 | CFSVMS | 73.77 | 79 | 74.73 | 8 | 8 |

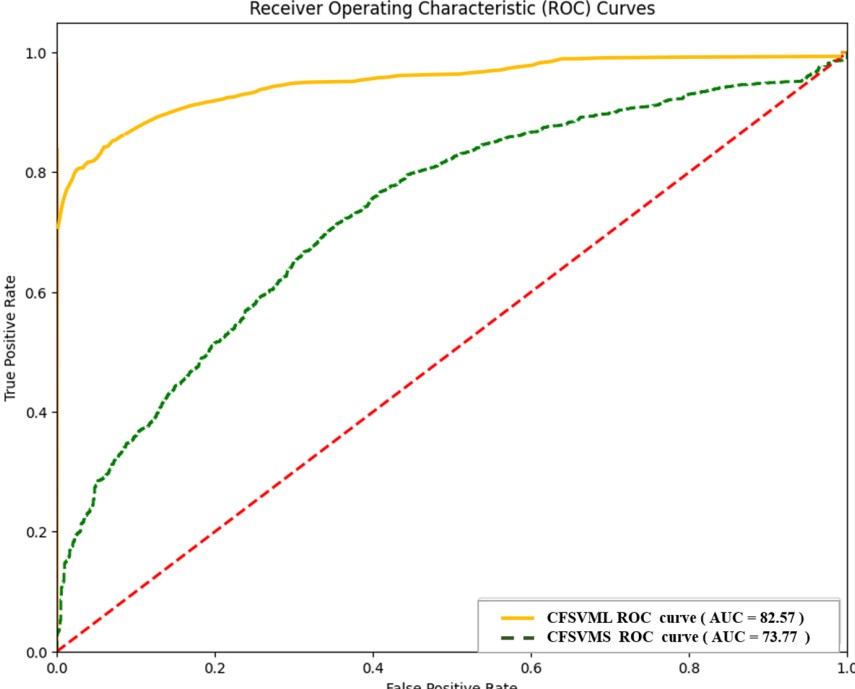

**Fig 12**. Comparative ROC plots of the best and the worst performing ML heart disease detection models for convolution features on ECG5000 dataset.

dataset. The three best-performing models in this analysis are AEFNN, AEFKNN and AEFRF each demonstrating notable accuracy levels. AEFNN achieved the highest accuracy at 97.96%, followed by AEFKNN with 93.73%, and AEFRF with 93.72%. Both ranking methods consistently ranked the same models as the top 1, 2, and 3. Fig 14 shows the Confusion metrics of three best-performing models from all 16 combinations of models on the MITBIH dataset.

Table 10 provides a performance comparison of the models on the ECG5000 dataset. The top three models identified are AEFSVMR, AEFSVEP, and AEFRF. The AEFSVMR and AEFSVEP models demonstrated similarly impressive performance, achieving an accuracy of 99.33%, an F1-score of 99.00, and an AUC of 99.33. The AEFRF model also performed exceptionally well, with an accuracy of 99.20%, an F1-score of 99.00, and an AUC of 99.22. Notably, a total of seven models achieved accuracy scores exceeding 98.00%. Both ranking methodologies employed yielded consistent rankings for the models. Fig 14 presents the confusion matrices for the three best-performing models among the 16 model combinations evaluated on the MITBIH dataset and ECG5000 dataset.

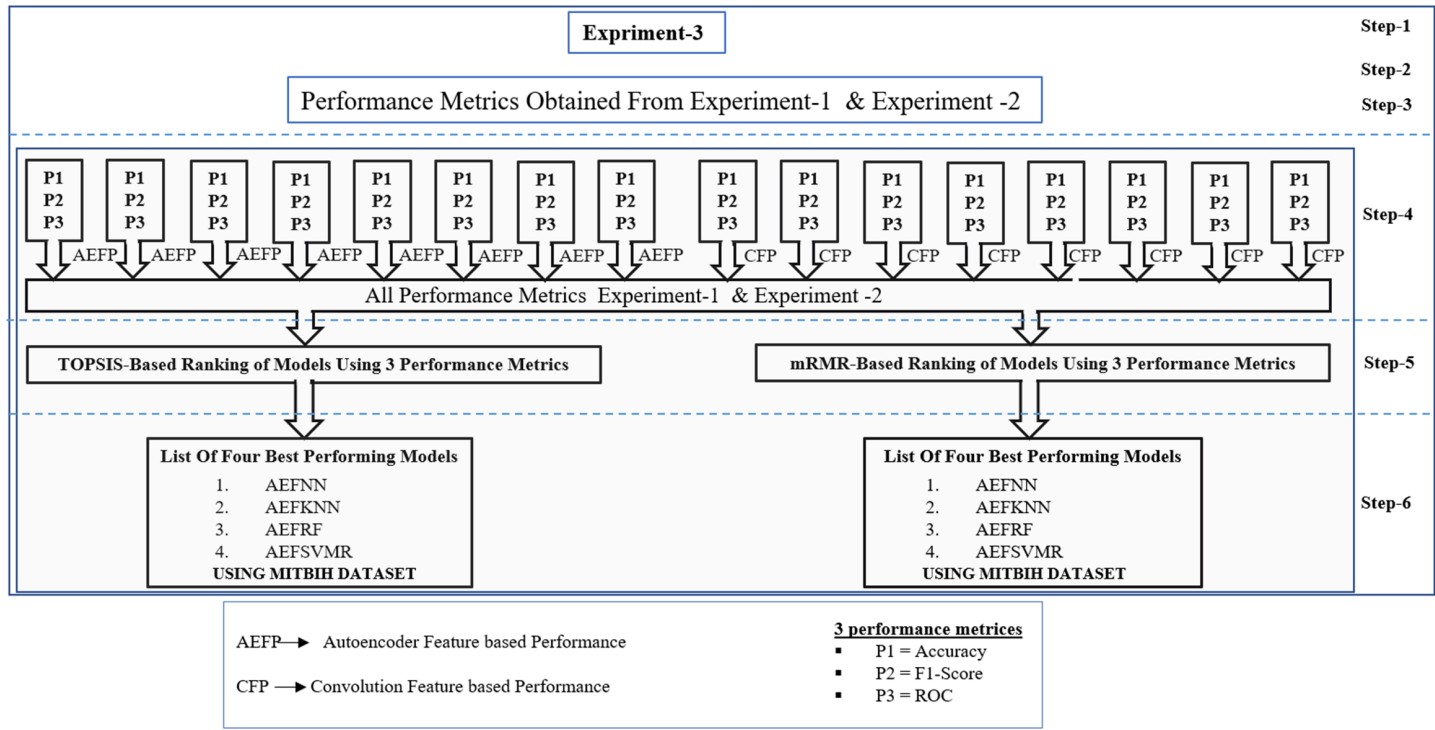

**Fig 13**. Systematic block cum flow diagram to identify best performing ML models for heart disease detection from ECG signal of patient.

**Table 9**. Overall evaluation of ranks of proposed ML models irrespective of two types of input features from the MITBIH dataset.

| SN | Method | Auc | f1-score | Accuracy | Rank(mRMR) | Rank(Topsis) | Statistical Rank |
|----|--------|-----|----------|----------|------------|--------------|------------------|
| 1 | AEFNN | 97.57 | 99.00 | 97.96 | 1 | 1 | 1 |
| 2 | AEFKNN | 92.67 | 96.00 | 93.72 | 2 | 3 | 2 |
| 3 | AEFRF | 91.11 | 96.00 | 93.72 | 3 | 3 | 2 |
| 4 | AEFSVMR | 90.12 | 95.00 | 92.27 | 4.5 | 5 | 3 |
| 5 | AEFSVMP | 90.12 | 95.00 | 92.27 | 4.5 | 5 | 3 |
| 6 | AEFDT | 83.37 | 90.00 | 86.34 | 6 | 6 | 4 |
| 7 | AESVML | 75.37 | 88.00 | 82.52 | 7 | 7 | 5 |
| 8 | CFNN | 75.11 | 86.00 | 79.42 | 8 | 8 | 6 |
| 9 | CFSVMR | 73.92 | 86.00 | 79.33 | 9.5 | 10 | 7 |
| 10 | CFSVMP | 73.92 | 86.00 | 79.33 | 9.5 | 10 | 7 |
| 11 | CSVML | 73.96 | 86.00 | 79.28 | 11 | 11 | 8 |
| 12 | CFSVMS | 68.90 | 83.00 | 75.28 | 12 | 12 | 9 |
| 13 | CFKNN | 69.66 | 84.00 | 72.21 | 13 | 13 | 10 |
| 14 | CFDT | 65.92 | 81.00 | 72.16 | 14 | 14 | 11 |
| 15 | CFRF | 65.88 | 81.00 | 72.14 | 15 | 15 | 12 |
| 16 | AEFSVMS | 65.15 | 80.00 | 71.75 | 16 | 16 | 13 |

## Insights from the three experiments

In this study, a comprehensive exploration of various hybrid models was conducted using the MIT-BIH and ECG5000 datasets. A total of 16 distinct algorithmic combinations were developed and implemented, incorporating two different ranking methodologies to enhance performance evaluation. Table 11 presents the overall ranking results obtained using two different datasets and two distinct ranking schemes. Our observations are:

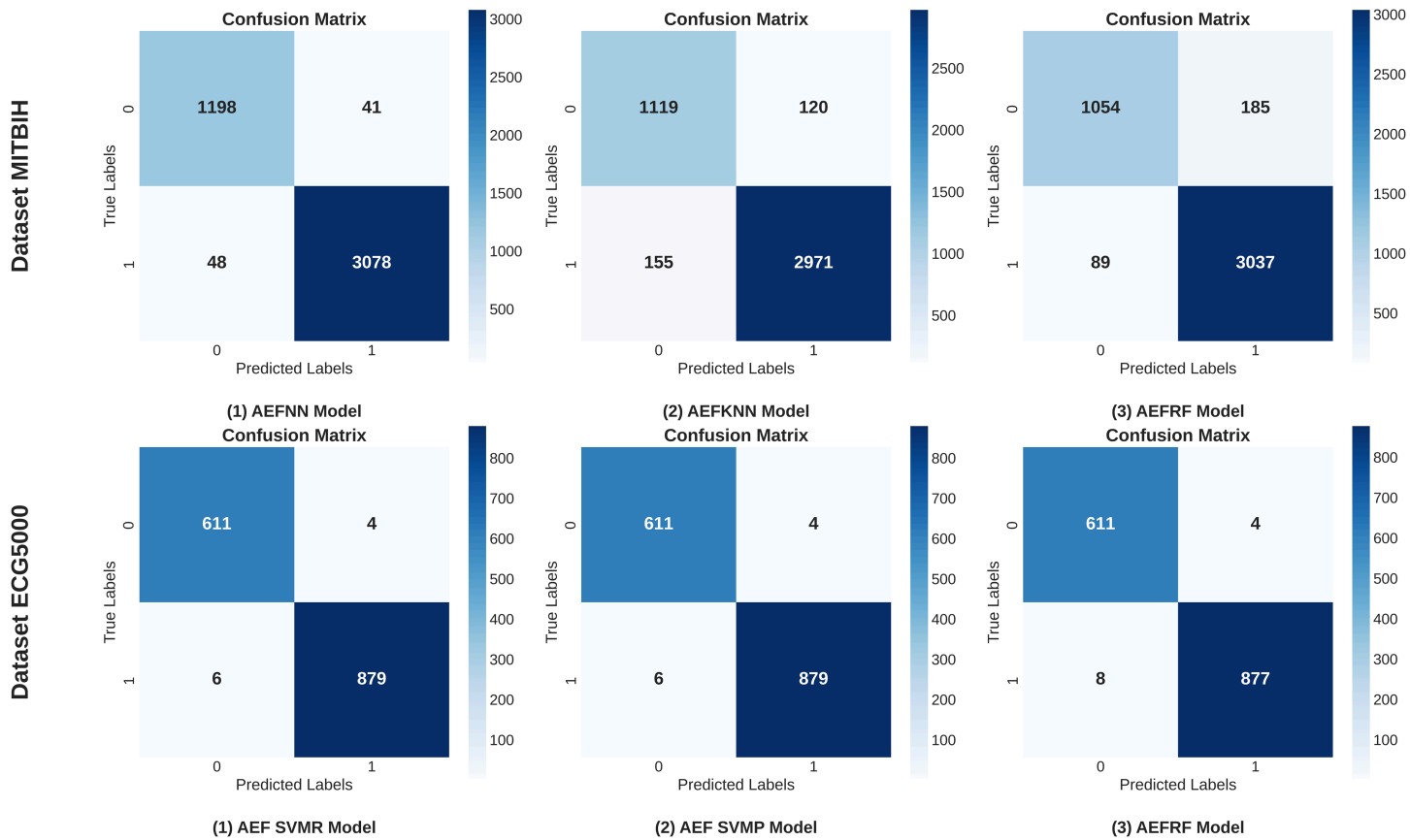

**Fig 14**. Top three performing models confusion matrix on MITBIH and ECG500 datasets.

**Table 10**. Overall evaluation of ranks of proposed ML models irrespective of two types of input features from the ECG5000 Dataset.

| SN | Method | Auc | f1-score | Accuracy | Rank(mRMR) | Rank(TOPSIS) | Statistical Rank |
|----|--------|-----|----------|----------|------------|--------------|------------------|
| 1 | AEFSVMR | 99.33 | 99.00 | 99.33 | 1.5 | 2 | 1 |
| 2 | AEFSVMP | 99.33 | 99.00 | 99.33 | 1.5 | 2 | 1 |
| 3 | AEFRF | 99.22 | 99.00 | 99.20 | 3 | 5 | 2 |
| 4 | AEFNN | 98.55 | 99.00 | 99.20 | 4.5 | 5 | 2 |
| 5 | AEFKNN | 98.55 | 99.00 | 99.20 | 4.5 | 5 | 2 |
| 6 | AEFSVML | 98.32 | 99.00 | 98.46 | 6 | 6 | 3 |
| 7 | AEFDT | 97.92 | 98.00 | 97.93 | 7 | 7 | 4 |
| 8 | AEFSVMS | 91.95 | 93.00 | 92.00 | 8 | 8 | 5 |
| 9 | CFSVML | 82.57 | 88.00 | 85.00 | 9 | 9 | 6 |
| 10 | CFSVMR | 81.97 | 88.00 | 84.66 | 11.5 | 11 | 7 |
| 11 | CFSVMP | 81.97 | 88.00 | 84.66 | 11.5 | 11 | 7 |
| 12 | CFNN | 82.33 | 88.00 | 84.39 | 10 | 12 | 8 |
| 13 | CFKNN | 80.50 | 86.00 | 77.73 | 13 | 13 | 9 |
| 14 | CFRF | 76.71 | 81.00 | 77.77 | 14.5 | 15 | 10 |
| 15 | CFDT | 76.71 | 81.00 | 77.77 | 14.5 | 15 | 10 |
| 16 | CFSVMS | 73.77 | 79.00 | 74.73 | 16 | 16 | 11 |

- Using Autoencoder Features (AEF), all classifiers achieved higher scores. Based on this experiment, AEF is demonstrated to be more effective than Conventional Features (CF).
- In this study neural network classifier performs higher than others on MITBIH dataset with autoencoder features.

**Table 11**. Model comparison between MITBIH and ECG5000 datasets.

| Dataset:- | MITBIH | | ECG5000 | |
|---|---|---|---|---|
| Feature:- | (CF) Feature | (AEF) Feature | (CF) Feature | (AEF) Feature |
| 1 | NN | NN | NN | SVMR |
| 2 | SVMR | KNN | SVMR | SVMP |
| 3 | SVMP | RF | SVMP | RF |

- The SVMR and SVMP models demonstrate comparable performance, even when assessed using different feature sets. This is illustrated in Table 9 and Table 10.
- The KNN and RF models demonstrate similar performance in this study, regardless of the two different feature sets used for evaluation.
- The hybrid model AEFNN demonstrates high performance across both datasets analyzed in this study.
- Ranking methods face conflict during critical conditions Table 10, Table 9.
- Table 11 shows the model comparison between MITBIH and ECG5000 datasets.

Table 9 and Table 10 show the Feature comparison between the MITBIH and ECG5000 datasets. In our study, we found that For ECG5000 and MITBIH datasets, Autoencoder-based Features are better than Convolution-based Features.

**Statistical analysis.** To assess whether the observed differences in model performance were statistically significant, McNemar's test [40] was conducted on the predictions of the top-performing models for both the MIT-BIH and ECG5000 datasets.

For the MIT-BIH dataset, five models achieved accuracies above 90%. However, AEFSVMR and AEFSVMP produced identical outputs in terms of accuracy, AUC, and F1-score (as reported in Table 9). Therefore, we considered the remaining four distinct models for pairwise comparisons. Similarly, for the ECG5000 dataset, six models achieved accuracies above 98%, but two pairs (AEFSVMR vs. AEFSVMP and AEFNN vs. AEFKNN) yielded identical results (as reported in Table 10), leaving four distinct models for comparison.

Since these comparisons involve multiple hypothesis tests (six pairwise comparisons per dataset), p-values were adjusted using the Holm–Bonferroni method to control the family-wise error rate at $\alpha = 0.05$. This approach provides a robust correction while retaining statistical power.

Table 12 presents the results for the MIT-BIH dataset. The upper triangle indicates whether the difference between a pair of models is statistically significant after Holm–Bonferroni correction (✓ = significant, ✗ = not significant), while the lower triangle reports the corresponding adjusted p-values. The results show that most comparisons involving AEFNN and AEFSVMR were significant, whereas the difference between AEFKNN and AEFRF was not. This suggests that AEFNN and AEFSVMR achieved significantly stronger performance than the other models.

Table 13 summarizes the results for the ECG5000 dataset. Here, only the comparison between AEFSVMR and AEFSVML remained statistically significant after correction, indicating a meaningful difference in predictive performance between these two models. All other model pairs showed no significant differences, suggesting that their performance was statistically indistinguishable despite numerical differences in accuracy.

Overall, the statistical analysis confirms that the superior performance of certain models—specifically AEFNN and AEFSVMR on MIT-BIH, and AEFSVMR on ECG5000—is not merely due to chance. These results provide rigorous evidence that the proposed models offer genuine and statistically validated improvements over competing approaches.

## Compare the result with other existing methods

The proposed method is compared with various existing methods. Table 14 showed the Comparison of the accuracy (%) of the proposed best-performing model with those obtained by other reported models using the dataset MIT-BIH,

**Table 12. Pairwise McNemar's test for Top 4 models on the ECG5000 dataset.** Upper triangle shows significance: "✓" indicates a statistically significant difference after Holm–Bonferroni correction ($\alpha = 0.05$), while "×" indicates no significant difference. Lower triangle shows Holm–Bonferroni adjusted p-values.

|  | AEFKNN | AEFRF | AEFSVML | AEFSVMR |
|---|---|---|---|---|
| AEFKNN | — | × | × | × |
| AEFRF | 0.3554 | — | × | × |
| AEFSVML | 0.908997 | 0.06363 | — | ✓ |
| AEFSVMR | 0.08984 | 0.908997 | 0.02655 | — |

**Table 13. Pairwise McNemar's Test for Top 4 Models on the MIT-BIH Dataset.** Upper triangle shows significance: "✓" indicates a statistically significant difference after Holm–Bonferroni correction ($\alpha = 0.05$), while "×" indicates no significant difference. Lower triangle shows Holm–Bonferroni adjusted p-values.

|  | AEFKNN | AEFNN | AEFRF | AEFSVMR |
|---|---|---|---|---|
| AEFKNN | — | ✓ | × | ✓ |
| AEFNN | 1.166e-36 | — | ✓ | ✓ |
| AEFRF | 0.9019 | 4.801e-41 | — | ✓ |
| AEFSVMR | 0.001889 | 5.326e-56 | 0.0006305 | — |

**Table 14. Comparison of the accuracy (%) of the proposed best-performing model with those obtained by other reported models using the dataset MIT-BIH.**

| Reference | Techniques | Accuracy(%) |
|---|---|---|
| Shin et al. [14] | AnoGAN | 94.75 |
| Sivapalan et al. [15] | LSTM / MLP-NN | 97.00 |
| Badawi et al. [26] | ANN | 99.10 |
| Zubair et al. [19] | CNN | 96.36 |
| Mandala et al. [20] | Decision Tree | 95.00 |
| Qin et al. [22] | GNN | 95.50 |
| **Proposed** | **AEFNN** | **97.96** |

**Table 15. Comparison of the accuracy (%) of the proposed best-performing model with those obtained by other reported models using the dataset ECG5000.**

| Reference | Techniques | Accuracy(%) |
|---|---|---|
| Chio et al. [27] | Inception V3 | 96.00 |
| Matias et al. [23] | VAE network | 98.79 |
| Alamr et al. [21] | AE + RNN , AE + LSTM | 89.5 |
| Xiu et al. [17] | TLVG | 93.02 |
| **proposed** | **AEFSVMR** | **99.33** |

and Table 15 showed Comparison of the accuracy (%) of the proposed best-performing model with those obtained by other reported models using the dataset ECG5000. Through an extensive combination of methodologies and rigorous research, we ultimately identified the four top-performing models: AEFNN, AEFKNN, AEFRF and AEFSVMR for the MIT-BIH dataset, and AEFSVMR, AEFSVMP, AEFRF and AEFRF for the ECG5000 dataset.

## Conclusion

The utilization of machine learning and deep learning techniques holds significant promise in revolutionizing the detection and classification of arrhythmias using ECG signals. Traditional methods, although effective, are often time-consuming and prone to errors due to the manual interpretation of simplistic algorithmic approaches. This paper specifically focuses on an in-depth comparative study of two feature extraction techniques, autoencoders and conventional convolution operation, particularly for feature extraction, and subsequently five machine learning techniques, such as NN, SVM, DT, KNN,

and RF, for arrhythmia detection using those extracted features. The performance of these models has been assessed and compared using two standard ECG datasets. The proposed methodology first involves training an autoencoder using ECG samples to extract essential features from the data. After successful training autoencoder is truncated at the bottleneck and then utilizes the trained encoder part, dimensionality reduction is achieved, facilitating the extraction of desired features. These features are subsequently used to train various machine learning classifiers. The total number of models developed is 16 which have been tested and various performance measures. Two different measures have been determined, and ranking algorithms have been employed to compare the overall performance of these models, which makes these instigations more efficacious. In essence, the integration of advanced machine learning and two types of features extracted from the ECG datasets offers a promising approach for obtaining the best possible accurate detection of arrhythmias from ECG signals, thereby contributing to improved diagnosis and treatment outcomes, ultimately saving lives affected by CD. After the assessment of the models investigated in this study, it was demonstrated that AEFNN, AEFKNN, and AEFRF achieved the 1st, 2nd, and 3rd ranks, respectively, among the 16 machine learning model combinations analyzed. Our study not only reports high performance metrics for the proposed models but also validates their superiority using McNemar's test. This statistical comparison confirms the significant improvement achieved by AEFNN and AEFSVMR over their counterparts on the MIT-BIH and ECG5000 datasets, respectively.

## Acknowledgments

This work was supported by the Zhejiang Provincial Natural Science Foundation (QN25H180024) and Zhejiang Provincial Health Commission Medical Health Science and Technology Project (Grant No. 2024KY1036).

## Author contributions

**Conceptualization:** Subir Biswas, Prabodh Kumar Sahoo.

**Data curation:** Subir Biswas.

**Formal analysis:** Subir Biswas, Adyasha Rath, Prince Jain, Haipeng Liu, Xinhong Wang.

**Funding acquisition:** Haipeng Liu, Xinhong Wang.

**Investigation:** Subir Biswas, Prabodh Kumar Sahoo, Adyasha Rath.

**Methodology:** Prabodh Kumar Sahoo, Adyasha Rath.

**Project administration:** Brajesh Kumar, Ganapati Panda, Xinhong Wang.

**Resources:** Adyasha Rath.

**Software:** Brajesh Kumar.

**Supervision:** Prabodh Kumar Sahoo, Prince Jain, Ganapati Panda.

**Validation:** Brajesh Kumar, Prince Jain, Ganapati Panda, Haipeng Liu.

**Visualization:** Subir Biswas, Brajesh Kumar, Prince Jain, Haipeng Liu, Xinhong Wang.

**Writing – original draft:** Subir Biswas.

**Writing – review & editing:** Brajesh Kumar, Adyasha Rath, Prince Jain, Haipeng Liu.

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
