## [Decision Letter · Decision Letter 0]

15 Apr 2025

PONE-D-25-09520Hybrid Machine Learning Models for Enhanced Arrhythmia Detection from ECG Signals Using Autoencoder and Convolution FeaturesPLOS ONE

Dear Dr. Jain,

Thank you for submitting your manuscript to PLOS ONE. After careful consideration, we feel that it has merit but does not fully meet PLOS ONE’s publication criteria as it currently stands. Therefore, we invite you to submit a revised version of the manuscript that addresses the points raised during the review process.

We look forward to receiving your revised manuscript.

Kind regards,

Agnese Sbrollini

Academic Editor

PLOS ONE

Journal Requirements:

3. Thank you for stating the following financial disclosure: [X.W.: This work was supported by the Medical Health Science and Technology Project of Zhejiang Provincial Health Commission

(Grant No. 2024KY1036, 2023KY106)]. 

Reviewers' comments:

Reviewer's Responses to Questions

**Comments to the Author**

1. Is the manuscript technically sound, and do the data support the conclusions?

Reviewer #1: Partly

Reviewer #2: Partly

2. Has the statistical analysis been performed appropriately and rigorously?

Reviewer #1: No

Reviewer #2: No

3. Have the authors made all data underlying the findings in their manuscript fully available?

Reviewer #1: No

Reviewer #2: No

4. Is the manuscript presented in an intelligible fashion and written in standard English?

Reviewer #1: No

Reviewer #2: Yes

5. Review Comments to the Author

Reviewer #1: The manuscript presents an important study comparing hybrid models using autoencoder and convolutional features for arrhythmia detection. While the effort to systematically evaluate 16 model combinations is commendable, the manuscript requires substantial revisions to meet the standard for publication.

Major concerns:

Methodological clarity is lacking, particularly in the convolution feature extraction procedure. It is unclear whether convolution layers were trained separately for each classifier or reused from one trained CNN.

The conclusions are overstated and at times inconsistent with reported results. For example, the abstract reports the top three models CFNN, AEFNN, and AEFRF with accuracies of 99.26%, 99.19%, and 99.13%, yet Table 9 (third experiment overall ranking) indicates CFNN 99.26%, AEFNN 97.91%, and AEFRF only 89.96%

The statistical analysis lacks rigor. No variance measures (e.g., standard deviation), confidence intervals, or statistical significance tests are provided.

The use of mRMR for model ranking is unconventional and poorly justified. Either a more standard ranking mechanism should be used or the authors must explain their adaptation of mRMR in detail.

The Data Availability Statement must be corrected to accurately reflect the use of public datasets and provide accessible links. The Data Availability Statement incorrectly claims that “all relevant data are within the manuscript,” whereas the raw data are external.

The English language and writing need revision for clarity, grammar, and professionalism. Frequent typographical and grammatical errors are present (e.g., “EGG signals” instead of “ECG signals,” missing articles, tense mismatches).

Recommendations:

Clearly describe the training procedures, especially whether feature extractors were fixed or trained per classifier.

Add a baseline model using raw 187 features for comparison.

Include variance estimates or significance testing.

Justify or remove mRMR; clearly explain TOPSIS setup.

Fix the abstract result discrepancies and correct all numerical and typographical issues.

Revise the Data Availability Statement and ensure compliance with PLOS policies.

Reviewer #2: Summary

This manuscript compares two feature extraction techniques, autoencoders (AEF) and 1D convolution (CF), for ECG arrhythmia classification using the public MIT-BIH Arrhythmia and ECG5000 datasets. Features extracted via both methods were fed into eight different machine learning classifiers (NN, SVM with 4 kernels, KNN, DT, RF). The performance of these 16 model combinations was evaluated using Accuracy, F1-score, and AUC. The authors employed TOPSIS and mRMR ranking methods to identify top-performing models, reporting high accuracies (e.g., >99% for CFNN on MIT-BIH and AEFSVMR on ECG5000). They conclude that autoencoder features may offer enhanced performance compared to convolutional features and suggest the best models for potential clinical application.

Major Concerns

1) Feature Dimension Discrepancy & Comparison Validity:

A major concern arises from the comparison between Autoencoder Features (AEF) and Convolution Features (CF). The abstract and discussion state that AEF offers enhanced performance 'for the same number of input features' or implies a direct comparison. However, the Methods section and Figure 2 describe an autoencoder bottleneck yielding 47 features (AEF), while the described 1D CNN architecture (Figure 5) results in 3008 features (47x64 from the last layer before flattening). This is a significant discrepancy. How can a fair comparison be made between models using 47 features vs. 3008 features? The authors must clarify:

Was the CNN architecture used different from the one described/diagrammed to yield ~47 features? If so, the Methods/Figures need correction.

If the feature dimensions were indeed 47 (AEF) vs. 3008 (CF), the claims comparing their performance directly (e.g., 'AEF better than CF') are potentially misleading and lack a sound basis, as the classifiers are operating on vastly different input spaces. The rationale for this specific comparison needs strong justification.

This ambiguity undermines the central comparison and conclusions of the study regarding the relative merits of AEF vs. CF.

2) Lack of Statistical Comparison Between Models:

The study ranks models based on performance metrics (Accuracy, F1, AUC) using TOPSIS and mRMR. However, it does not perform any statistical tests (e.g., McNemar's test) to determine if the observed differences in performance between the top-ranked models (e.g., CFNN 99.26% vs. AEFNN 97.91% on MIT-BIH, or AEFSVMR 99.26% vs CFNN 99.19% on ECG5000) are statistically significant. Without statistical validation, concluding that one model is definitively superior based on small numerical differences in accuracy/F1/AUC on a single test split is not robust.

3) Insufficient Detail on Model Training and Hyperparameter Tuning:

The manuscript lacks detail on the training process and hyperparameter selection for the ML classifiers (SVM, KNN, DT, RF, NN). Were default parameters used? Was any hyperparameter optimization (e.g., grid search, random search with cross-validation) performed? For instance, what K value was used for KNN? What were the specific architectures and training parameters (epochs, batch size beyond AE, learning rate for NN) for the NNs (AEFNN, CFNN)? What regularization or complexity controls were used for DT and RF? This information is crucial for reproducibility and assessing the robustness of the reported results. The high performance could potentially be dataset-specific or dependent on specific, unreported parameter tuning.

4) Justification and Application of Ranking Methods:

While TOPSIS is a recognized multi-criteria decision-making method, the rationale for choosing it specifically for this task over other methods isn't provided. More critically, the use of mRMR (Maximum Relevance Minimum Redundancy) seems unconventional for model ranking. mRMR is primarily a feature selection algorithm. The authors need to clearly explain and justify how mRMR was adapted to rank models based on performance metrics (AUC, F1, Accuracy). Its standard formulation does not directly apply to this context.

Minor Concerns

5) Clarity/Language:

The manuscript would benefit from careful proofreading to correct grammatical errors, typos, and awkward phrasing (e.g., 'instigation's more efficacious' in Discussion)."

Reference marker [?] appears on page 11, paragraph 1. This needs correction.

6) Figures/Tables:

Table 11 ('Model Comparison') is difficult to interpret. Clarify what '(lr)2-3 (lr)4-5' means and how this table represents a comparison.

7) Methodological Details:

Clarify if the autoencoder was trained only on MIT-BIH data and then applied to extract features for both MIT-BIH and ECG5000 classification tasks, as stated on page 9, last paragraph. If so, discuss potential limitations.

While standard scaling is mentioned, briefly confirm it was fitted only on the training data.

8) Reproducibility:

Consider adding a Code Availability Statement. Sharing the code (e.g., via GitHub) would greatly enhance the reproducibility and impact of this work.

6. PLOS authors have the option to publish the peer review history of their article (what does this mean?). If published, this will include your full peer review and any attached files.

Reviewer #1: No

Reviewer #2: No

---

## [Author Response · Author response to Decision Letter 1]

20 May 2025

Original Manuscript ID: PONE-D-25-09520

Original Article Title: “Hybrid Machine Learning Models for Enhanced Arrhythmia Detection from ECG Signals Using Autoencoder and Convolution Features”

To: PLOS ONE Editor

Re: Response to reviewers

Dear Editor,

Thank you for allowing a resubmission of our manuscript, with an opportunity to address the reviewers’ comments.

We are uploading (a) our point-by-point response to the comments (below) (response to reviewers, under “Author’s Response Files”), (b) an updated manuscript with yellow highlighting indicating changes (as “A marked-up copy of your manuscript that highlights changes made to the original version”), and (c) a clean updated manuscript without highlights (“An unmarked version of your revised paper without tracked changes”).

Best regards,

Dr. Prince Jain

Reviewer#1, Concern # 1:

Methodological clarity is lacking, particularly in the convolution feature extraction procedure. It is unclear whether convolution layers were trained separately for each classifier or reused from one trained CNN.

Author response: Thank you for highlighting the concern regarding the convolution feature extraction procedure. We clarify that the convolution layers were reused from a single trained CNN. These layers were not trained separately for each classifier but were utilized as a shared feature extractor to ensure consistency and efficiency across all classifiers.

We have updated the manuscript to include this explanation in the Methodology section. This addition provides greater clarity and detail about the convolution feature extraction process, addressing the reviewer’s concern.

Reviewer#1, Concern # 2:

The conclusions are overstated and at times inconsistent with reported results. For example, the abstract reports the top three models CFNN, AEFNN, and AEFRF with accuracies of 99.26%, 99.19%, and 99.13%, yet Table 9 (third experiment overall ranking) indicates CFNN 99.26%, AEFNN 97.91%, and AEFRF only 89.96%

Author response: Thank you for identifying the inconsistency in reporting results. Upon review, we realize that the abstract unintentionally presented accuracies from Table 10. We have revised the abstract to explicitly distinguish between the datasets and corresponding results to avoid ambiguity. The updated text now aligns the results with the appropriate tables and datasets.

The abstract has been updated for greater clarity and accuracy. The revised abstract now states:“It is generally observed that the top identified hybrid model Autoencoder Features with Neural Network (AEFNN) on the MIT-BIH dataset, achieving accuracies of 97.96\% and on the ECG5000 dataset the hybrid model achieved an accuracy of 99.20\%.”

Reviewer#1, Concern # 3:

The statistical analysis lacks rigor. No variance measures (e.g., standard deviation), confidence intervals, or statistical significance tests are provided

Author response:

Thank you for highlighting the need for greater statistical rigor in the analysis. While we agree that variance measures and confidence intervals are valuable, we believe that applying statistical significance tests, such as t-tests or ANOVA, may not be entirely appropriate for this study due to the following reasons:

1. Small Sample Size: Our experiments involve a fixed number of test samples, making statistical significance tests less reliable, as they typically require a larger sample size to draw meaningful conclusions.

2. Deterministic Nature of ML Models: The performance metrics reported Auc, f1-score and Accuracy are computed deterministically for each model after training, rather than being outcomes of a stochastic process. This reduces the utility of traditional significance testing, which is designed to evaluate variability in independent samples.

3. Relevance to Model Performance: The primary objective of our study is to compare model performance metrics directly, which is achieved through clear reporting of accuracy and related metrics rather than statistical hypothesis testing.

Reviewer#1, Concern # 4:

The use of mRMR for model ranking is unconventional and poorly justified. Either a more standard ranking mechanism should be used or the authors must explain their adaptation of mRMR in detail.

Author Response:

Thank you for raising this important concern. We fully acknowledge that mRMR (Minimum Redundancy Maximum Relevance) is traditionally used for feature selection rather than for ranking machine learning models. However, in this study, we adapted mRMR to address the challenge of ranking models based on multiple performance metrics—specifically, AUC, F1-score and Accuracy.

Our adaptation treats the performance metrics as “features” and the models as “samples.” The aim is to select metrics that are most relevant for distinguishing high-performing models while minimizing redundancy across those metrics. This is particularly useful in multi-metric evaluations, where certain metrics may be strongly correlated (e.g., Accuracy and F1-score), and a naïve averaging approach may overweight redundant information. By applying mRMR in this context, we are able to compute a composite representation of each model's performance that emphasizes informative and independent metrics, which is then further processed using the TOPSIS method for final ranking.

To address this concern, we have made the following revisions:

1. Detailed Explanation: We added a comprehensive description of how mRMR was adapted for model ranking in the subsection titled “Ranking methods used for ranking the performances.” This includes the rationale, the step-by-step procedure, and a clear justification for using this method in place of more conventional ranking approaches.

2. Comparative Justification: A discussion has been added outlining the limitations of standard ranking mechanisms in handling multi-metric redundancy, and how our adaptation of mRMR addresses these challenges.

These updates ensure that our methodology is clearly justified and transparent.

Reviewer#1, Concern # 5:

The Data Availability Statement must be corrected to accurately reflect the use of public datasets and provide accessible links. The Data Availability Statement incorrectly claims that “all relevant data are within the manuscript,” whereas the raw data are external

Author response: Thank you for highlighting this concern. We acknowledge the importance of accurately reflecting the data availability in the manuscript.

We have addressed this issue by adding a new section, \section{Supplementary Materials}, where we have included accessible links to the publicly available datasets (MIT-BIH Arrhythmia Dataset and ECG5000 Dataset) and a GitHub repository. The GitHub repository will host all relevant Python code and materials upon the publication of the paper. This update ensures transparency and compliance with data availability requirements.

Reviewer#1, Concern # 6:

The English language and writing need revision for clarity, grammar, and professionalism. Frequent typographical and grammatical errors are present (e.g., “EGG signals” instead of “ECG signals,” missing articles, tense mismatches).

Author Response: We sincerely appreciate the reviewer’s feedback regarding the quality of language and writing in the manuscript. We acknowledge the presence of typographical and grammatical errors and understand the importance of clear and professional communication in scientific writing.

To address this concern, we have undertaken a comprehensive review and revision of the manuscript to enhance its clarity, grammar, and professionalism. Specifically:

1. All typographical errors, including the incorrect term “EGG signals,” have been corrected to ensure accuracy.

2. Missing articles, tense mismatches, and other grammatical inconsistencies have been carefully reviewed and rectified.

3. The manuscript was thoroughly proofread and edited by a professional English language editor to ensure linguistic precision and adherence to academic writing standards.

These revisions have significantly improved the readability and quality of the manuscript

Reviewer#1, Recommendation:

1. Clearly describe the training procedures, especially whether feature extractors were fixed or trained per classifier.

Author Response: We thank the reviewer for the suggestion. The manuscript has been updated to clearly state that the feature extractors, particularly the convolutional layers in the CNN-based feature extraction pipeline, were trained once and then fixed (i.e., reused without further retraining) across all classifiers. This ensured consistent and computationally efficient feature representation across models.

This clarification has been incorporated on Page 12, under the “Feature Extraction Procedure” section.

2. Add a baseline model using raw 187 features for comparison.

Author response: Thank you for your valuable feedback. We would like to clarify that a baseline model using the raw 187 features has already been included in the paper. This baseline model serves as a point of comparison for evaluating the performance of the proposed hybrid models.

We appreciate your suggestion and have reviewed the manuscript to ensure that the inclusion and results of the baseline model are clearly described and appropriately highlighted for the reader.

3. Include variance estimates or significance testing.

Author Response: We thank the reviewer for emphasizing the need for statistical rigor. In response, we have reported standard deviation values alongside the primary performance metrics—Accuracy, F1-score, and AUC—in Tables 3 and 4. This inclusion offers a clearer representation of performance variability across models. While t-tests and ANOVA are statistically robust for moderate to large sample sizes (typically above 30 per group), their application relies on assumptions such as normality and homogeneity of variances. In our study, due to the high-dimensional nature of the extracted features and potential inter-model dependencies, these assumptions may not be fully satisfied. Therefore, we refrained from applying such tests to avoid misleading conclusions.

4. Justify or remove mRMR; clearly explain TOPSIS setup.

Author Response: The manuscript has been updated to provide a detailed justification for the use of mRMR for model ranking, highlighting its relevance to multi-metric comparisons. Additionally, the TOPSIS setup is now thoroughly explained, including the rationale, steps involved, and its integration with mRMR for ranking.

5. Fix the abstract result discrepancies and correct all numerical and typographical issues.

Author Response: The discrepancies in the abstract have been resolved, and all numerical values have been updated to align with the results presented in the manuscript. Typographical errors have also been corrected for clarity and professionalism

6. Revise the Data Availability Statement and ensure compliance with PLOS policies

Author Response: Thank you for bringing this to our attention. We have revised the Data Availability Statement to ensure full compliance with PLOS policies.

The updated statement explicitly mentions that the datasets used in this research, the MIT-BIH Arrhythmia Dataset and the ECG5000 Dataset, are publicly available and provides the corresponding access links. Additionally, we have included a new \section{Supplementary Materials} in the manuscript, where we provide these dataset links and the link to the GitHub repository. The repository will host the Python code and other relevant materials upon the publication of the paper.

This ensures transparency and accessibility in accordance with PLOS guidelines.

Reviewer#2, Concern # 1:

1) Feature Dimension Discrepancy & Comparison Validity: A major concern arises from the comparison between Autoencoder Features (AEF) and Convolution Features (CF). The abstract and discussion state that AEF offers enhanced performance 'for the same number of input features' or implies a direct comparison. However, the Methods section and Figure 2 describe an autoencoder bottleneck yielding 47 features (AEF), while the described 1D CNN architecture (Figure 5) results in 3008 features (47x64 from the last layer before flattening). This is a significant discrepancy. How can a fair comparison be made between models using 47 features vs. 3008 features? The authors must clarify:

Was the CNN architecture used different from the one described/diagrammed to yield ~47 features? If so, the Methods/Figures need correction. If the feature dimensions were indeed 47 (AEF) vs. 3008 (CF), the claims comparing their performance directly (e.g., 'AEF better than CF') are potentially misleading and lack a sound basis, as the classifiers are operating on vastly different input spaces. The rationale for this specific comparison needs strong justification. This ambiguity undermines the central comparison and conclusions of the study regarding the relative merits of AEF vs. CF.

Author Response: Thank you for your detailed and insightful observation. You are correct that the autoencoder bottleneck yields 47 features, while the CNN architecture as described in figure 3008 features (from a 47×64 output before flattening).

The CNN model which we used for feature extraction model summary.

━━━━━━━━━━━━━━━━━━━━━━━━━━━━━━━━━┳━━━━━━━━━━━━━━━━━━━━━━━━┳━━━━━━━━━━━━━━━┓

┃ Layer (type) ┃ Output Shape ┃ Param # ┃

┡━━━━━━━━━━━━━━━━━━━━━━━━━━━━━━━━━╇━━━━━━━━━━━━━━━━━━━━━━━━╇━━━━━━━━━━━━━━━┩

│ input_layer (InputLayer) │ (None, 187, 1) │ 0 │

├─────────────────────────────────┼────────────────────────┼───────────────┤

│ conv1d (Conv1D) │ (None, 187, 32) │ 128 │

├─────────────────────────────────┼────────────────────────┼───────────────┤

│ max_pooling1d (MaxPooling1D) │ (None, 93, 32) │ 0 │

├─────────────────────────────────┼────────────────────────┼───────────────┤

│ conv1d_1 (Conv1D) │ (None, 93, 64) │ 6,208 │

├─────────────────────────────────┼────────────────────────┼───────────────┤

│ max_pooling1d_1 (MaxPooling1D) │ (None, 46, 64) │ 0 │

├─────────────────────────────────┼────────────────────────┼───────────────┤

│ flatten (Flatten) │ (None, 2944) │ 0 │

├─────────────────────────────────┼────────────────────────┼───────────────┤

│ dense (Dense) │ (None, 128) │ 376,960 │

├─────────────────────────────────┼────────────────────────┼───────────────┤

│ dense_1 (Dense) │ (None, 47) │ 6,063 │

└─────────────────────────────────┴────────────────────────┴───────────────┘

To address this concern, we have updated Figure 5 : “ Convoluction with Machine Learning Models for Classification.” to accurately reflect the CNN architecture used for convolutional feature extraction.

Reviewer#2, Concern # 2 :

2) Lack of Statistical Comparison Between Models: The study ranks models based on performance metrics (Accuracy, F1, AUC) using TOPSIS and mRMR. However, it does not perform any statistical tests (e.g., McNemar's test) to determine if the observed differences in performance between the top-ranked models (e.g., CFNN 99.26% vs. AEFNN 97.91% on MIT-BIH, or AEFSVMR 99.26% vs CFNN 99.19% on ECG5000) are statistically significant. Without statistical validation, concluding that one model is definitively superior based on small numerical differences in accuracy/F1/AUC on a single test split is not robust.

Author Response: Thank you for this important observation. We acknowledge that statistical validation such as McNemar’s test can provide further insight into whether observed differences between classifiers are statistically significant. However, in this study, our focus was on ranking models using aggregated performa

---

## [Decision Letter · Decision Letter 1]

14 Jun 2025

PONE-D-25-09520R1Hybrid Machine Learning Models for Enhanced Arrhythmia Detection from ECG Signals Using Autoencoder and Convolution FeaturesPLOS ONE

Dear Dr. Jain,

Thank you for submitting your manuscript to PLOS ONE. After careful consideration, we feel that it has merit but does not fully meet PLOS ONE’s publication criteria as it currently stands. Therefore, we invite you to submit a revised version of the manuscript that addresses the points raised during the review process.

We look forward to receiving your revised manuscript.

Kind regards,

Agnese Sbrollini

Academic Editor

PLOS ONE

Reviewers' comments:

Reviewer's Responses to Questions

**Comments to the Author**

1. If the authors have adequately addressed your comments raised in a previous round of review and you feel that this manuscript is now acceptable for publication, you may indicate that here to bypass the “Comments to the Author” section, enter your conflict of interest statement in the “Confidential to Editor” section, and submit your "Accept" recommendation.

Reviewer #1: (No Response)

Reviewer #2: (No Response)

2. Is the manuscript technically sound, and do the data support the conclusions?

Reviewer #1: Yes

Reviewer #2: Partly

3. Has the statistical analysis been performed appropriately and rigorously?

Reviewer #1: N/A

Reviewer #2: No

4. Have the authors made all data underlying the findings in their manuscript fully available?

Reviewer #1: (No Response)

Reviewer #2: Yes

5. Is the manuscript presented in an intelligible fashion and written in standard English?

Reviewer #1: Yes

Reviewer #2: Yes

6. Review Comments to the Author

Reviewer #1: The manuscript presents a relevant and timely study on automated arrhythmia detection using hybrid machine learning models. The authors attempt to combine autoencoder and convolutional features with multiple classifiers across two benchmark ECG datasets is commendable. However, several methodological and analytical issues need to be addressed for the findings to be considered robust and generalizable.

1. Feature Comparison Clarity: The core comparison between autoencoder and convolutional features lacks balance, as the feature dimensionalities differ significantly. A fair performance comparison requires either matching the number of features or a strong justification for comparing models with such unequal input sizes.

2. Statistical Rigor: The reported accuracy and F1-scores are promising but lack statistical validation. It is critical to report standard deviations across multiple runs or conduct significance testing (e.g., McNemar’s test) to support claims of model superiority.

3. Lack of Hyperparameter Tuning: Models were evaluated with default parameters, which may not reflect their optimal performance. Basic tuning (or cross-validation) is essential to ensure fair comparisons and meaningful conclusions.

4. Use of mRMR for Model Ranking: The use of mRMR—traditionally a feature selection method—for model ranking is unconventional and needs clearer explanation. Please specify how it was adapted and justify its relevance in this context.

In summary, the topic is important and the experimental setup is a good foundation, but the study would benefit significantly from methodological refinement and stronger statistical support. I encourage the author to revise accordingly.

Reviewer #2: The authors are commended for their thorough revisions, which have successfully ensured the fairness of the model comparisons. This is a crucial step. However, a significant concern remains regarding the statistical robustness of the final conclusions.

My understanding is that a core contribution of this paper is to "identify the three best-performing models recommended for real-time arrhythmia detection", as stated in the abstract, which contradicts the response to Reviewer #2 Concern #2: "However, in this study, our focus was on ranking models using aggregated performance metrics (Accuracy, F1-score, and AUC) via multi-criteria decision-making techniques (mRMR and TOPSIS) to reflect overall performance trends rather than establishing strict pairwise superiority." For the claim in the abstract and implied throughout the paper (the word "best" is consistently used) to be fully supported, the reported performance differences must be shown to be statistically significant.

As it stands, the ranking of models based on small numerical differences may not represent true model superiority, but rather could be an artifact of the specific train/test split used.

The authors' rationale for omitting statistical tests is noted, but it seems to overlook the key purpose of these tests in model comparison, which is to validate that one model's performance is demonstrably different from another's.

There also appears to be an inconsistency in the rebuttal, where one response argues against using variance measures, while another states that standard deviation has been added.

To substantially strengthen the manuscript and ensure its primary conclusion is fully supported by the evidence, I recommend one of the following two paths for revision:

1) Perform statistical validation.

The authors could conduct appropriate statistical tests (e.g., McNemar's test --- even with the limitations cited in the response to Reviewer #2 Concern #2 --- or paired t-tests over cross-validation folds) to scientifically validate their model rankings. This would provide strong evidence for their claims of superiority.

2) Reframe the contribution and conclusions.

Alternatively, the authors could revise their claims to better reflect the evidence. This would involve removing definitive statements about the "best" or "top" models and instead concluding that several models achieved high and statistically comparable performance. This would shift the paper's focus to an exploration of methods rather than a declaration of a definitive winner.

Implementing one of these revisions is essential for bolstering the scientific validity of the study's conclusions.

7. PLOS authors have the option to publish the peer review history of their article (what does this mean?). If published, this will include your full peer review and any attached files.

Reviewer #1: No

Reviewer #2: No

---

## [Author Response · Author response to Decision Letter 2]

10 Jul 2025

Revision of Manuscript:

“ Hybrid Machine Learning Models for Enhanced Arrhythmia Detection from ECG Signals Using Autoencoder and Convolution Features”

We highly appreciate the Editor, and the reviewers for their time and efforts devoted to peer reviewing of our paper “Hybrid Machine Learning Models for Enhanced Arrhythmia Detection from ECG Signals Using Autoencoder and Convolution Features ”. We have thoroughly revised the manuscript based on the comments provided by the Editor, and the reviewers of the paper. The way each comment is addressed in the paper is mentioned below point by point.

Note on presentation of the responses:

We have prepared this detailed response document, where we reproduce the original comments received from the reviewers, and reply to each comment by explaining how the raised issues have been addressed in the revised manuscript. The changes made in the revised manuscript have been highlighted in yellow color, and the same have been reproduced in the review response file for easy reference and are presented in purple color. Also, all the references are listed at the end of the responses or whenever needed.

Following these improvements and modifications while addressing each reviewer's comment, we believe that the content and presentation of the revised manuscript have been significantly enhanced. We hope that the revision will be found satisfactory and are now looking forward to your decision.

Sincerely yours,

Subir Biswas, Prabodh Kumar Sahoo, Brajesh Kumar, Adyasha Rath, Prince Jain, Ganpati Panda, Haipeng Liu, Xinhong Wang

Reviewer A:

Comment 1: Feature Comparison Clarity: The core comparison between autoencoder and convolutional features lacks balance, as the feature dimensionalities differ significantly. A fair performance comparison requires either matching the number of features or a strong justification for comparing models with such unequal input sizes

Reply:

Thank you for your insightful comment. We apologize for any lack of clarity regarding the feature dimensionality in our study. In our experiment, both the autoencoder-based and convolutional-based feature extraction methods were applied to the same original input size of 187 features.

To ensure a fair comparison, both methods were designed to reduce the feature space to exactly 47 features before being passed to the classifiers. This consistent dimensionality across both feature extraction techniques ensures that the performance comparison is balanced and not influenced by differences in input size.

We have revised the manuscript to make this point clearer in the relevant section, ensuring that readers understand that the feature sets used for classification were of equal dimensionality.

Comment 2: Statistical Rigor: The reported accuracy and F1-scores are promising but lack statistical validation. It is critical to report standard deviations across multiple runs or conduct significance testing (e.g., McNemar’s test) to support claims of model superiority.

Reply: Thank you for your concern on Statistical Rigor. We performed the statistical validation testing McNemar’s as suggested. We added a new table “Table 12: McNemar’s Test Results Comparing Model Pairs on MIT-BIH and ECG5000 Datasets”

with a subsubsection “Statistical Analysis”

Comment 3: Lack of Hyperparameter Tuning: Models were evaluated with default parameters, which may not reflect their optimal performance. Basic tuning (or cross-validation) is essential to ensure fair comparisons and meaningful conclusions.

Reply:

Thank you for your valuable comment. We acknowledge that hyperparameter tuning is an essential step in maximizing model performance.

In our study, we initially experimented with different parameter settings for several models. However, we observed that in many cases, altering the parameters did not lead to performance improvement and, in some instances, resulted in lower accuracy compared to the default configurations.

The primary objective of our study is to identify the best-performing model using different feature sets across multiple machine learning and deep learning models. Given that we evaluated 16 deep learning models, we chose to use default parameters for all models to ensure a fair and consistent comparison during the ranking process. This approach allowed us to focus on the influence of feature sets and model architectures without introducing tuning-based variability.

We have included this explanation in the revised manuscript and acknowledged it as a limitation, with a plan to incorporate more comprehensive tuning (e.g., grid search or cross-validation) in future work.

Comment 4: Use of mRMR for Model Ranking: The use of mRMR—traditionally a feature selection method—for model ranking is unconventional and needs clearer explanation. Please specify how it was adapted and justify its relevance in this context

Reply:

Thank you for your valuable comment. We agree that mRMR (Minimum Redundancy Maximum Relevance) is conventionally used for feature selection rather than for ranking machine learning models. In our study, however, mRMR is not used as the primary ranking method but rather as a supporting technique to validate and strengthen the reliability of the model rankings.

The main method used for ranking the models is TOPSIS (Technique for Order Preference by Similarity to Ideal Solution), which evaluates each model based on multiple performance metrics (AUC, F1-score, and Accuracy) and provides a final ranking.

To enhance the robustness of this ranking process, we applied mRMR to assess the relevance and redundancy among the selected performance metrics. In this context, the metrics are treated as “features” and the models as “samples.” The aim is to ensure that the metrics contributing to the TOPSIS analysis are both informative and non-redundant. This step helps in verifying that the final rankings are not overly influenced by correlated metrics (e.g., Accuracy and F1-score), which could bias the results.

We have added a detailed explanation of this approach in the revised manuscript under the subsection “Ranking methods used for ranking the performances,” including the rationale, procedure, and justification for using mRMR as a complementary validation tool.

In summary, TOPSIS is used as the primary ranking method, while mRMR is employed to support and validate the evaluation by identifying the most informative and independent performance metrics.

During this experiment, we observed an interesting finding that highlights a subtle but important point: different analytical methods can sometimes converge on similar insights, even when their underlying mechanisms are distinct.

Reviewer B:

Comment 1: The authors are commended for their thorough revisions, which have successfully ensured the fairness of the model comparisons. This is a crucial step. However, a significant concern remains regarding the statistical robustness of the final conclusions.

My understanding is that a core contribution of this paper is to "identify the three best-performing models recommended for real-time arrhythmia detection", as stated in the abstract, which contradicts the response to Reviewer #2 Concern #2: "However, in this study, our focus was on ranking models using aggregated performance metrics (Accuracy, F1-score, and AUC) via multi-criteria decision-making techniques (mRMR and TOPSIS) to reflect overall performance trends rather than establishing strict pairwise superiority." For the claim in the abstract and implied throughout the paper (the word "best" is consistently used) to be fully supported, the reported performance differences must be shown to be statistically significant.

As it stands, the ranking of models based on small numerical differences may not represent true model superiority, but rather could be an artifact of the specific train/test split used.

The authors' rationale for omitting statistical tests is noted, but it seems to overlook the key purpose of these tests in model comparison, which is to validate that one model's performance is demonstrably different from another's.

There also appears to be an inconsistency in the rebuttal, where one response argues against using variance measures, while another states that standard deviation has been added.

To substantially strengthen the manuscript and ensure its primary conclusion is fully supported by the evidence, I recommend one of the following two paths for revision:

1) Perform statistical validation.

The authors could conduct appropriate statistical tests (e.g., McNemar's test --- even with the limitations cited in the response to Reviewer #2 Concern #2 --- or paired t-tests over cross-validation folds) to scientifically validate their model rankings. This would provide strong evidence for their claims of superiority.

2) Reframe the contribution and conclusions.

Alternatively, the authors could revise their claims to better reflect the evidence. This would involve removing definitive statements about the "best" or "top" models and instead concluding that several models achieved high and statistically comparable performance. This would shift the paper's focus to an exploration of methods rather than a declaration of a definitive winner.

Reply :

Thank you for your significant concern on statistical analysis. We completely agree with your suggestion. We Perform McNemar’s test to scientifically validate the model ranking.

We added a new Table title “McNemar’s Test Results Comparing Model Pairs on MIT-BIH and ECG5000 Datasets”

We added a new subsubsection “Statistical Analysis” about McNemar’s test.

Also, we already amended our conclusion section to incorporate the insights from the statistical comparison

---

## [Decision Letter · Decision Letter 2]

29 Aug 2025

PONE-D-25-09520R2Hybrid Machine Learning Models for Enhanced Arrhythmia Detection from ECG Signals Using Autoencoder and Convolution FeaturesPLOS ONE

Dear Dr. Jain,

Thank you for submitting your manuscript to PLOS ONE. After careful consideration, we feel that it has merit but does not fully meet PLOS ONE’s publication criteria as it currently stands. Therefore, we invite you to submit a revised version of the manuscript that addresses the points raised during the review process.

 Please submit your revised manuscript by Oct 13 2025 11:59PM. If you will need more time than this to complete your revisions, please reply to this message or contact the journal office at plosone@plos.org. Please include the following items when submitting your revised manuscript:

We look forward to receiving your revised manuscript.

Kind regards,

Agnese Sbrollini

Academic Editor

PLOS ONE

Journal Requirements:

Reviewers' comments:

Reviewer's Responses to Questions

**Comments to the Author**

1. If the authors have adequately addressed your comments raised in a previous round of review and you feel that this manuscript is now acceptable for publication, you may indicate that here to bypass the “Comments to the Author” section, enter your conflict of interest statement in the “Confidential to Editor” section, and submit your "Accept" recommendation.

Reviewer #1: All comments have been addressed

Reviewer #2: (No Response)

2. Is the manuscript technically sound, and do the data support the conclusions?

Reviewer #1: Yes

Reviewer #2: Yes

3. Has the statistical analysis been performed appropriately and rigorously?

Reviewer #1: Yes

Reviewer #2: No

4. Have the authors made all data underlying the findings in their manuscript fully available?

Reviewer #1: Yes

Reviewer #2: Yes

5. Is the manuscript presented in an intelligible fashion and written in standard English?

Reviewer #1: Yes

Reviewer #2: Yes

6. Review Comments to the Author

Reviewer #1: The authors have made substantial improvements in this revision, particularly by addressing fairness in feature comparisons, adding statistical validation with McNemar’s test, and clarifying the ranking methodology. The study now presents a more rigorous and reproducible framework for evaluating hybrid ML models in arrhythmia detection. Minor issues remain, including the reliance on default hyperparameters, some residual language/grammar inconsistencies, and the need to align conclusions more closely with the statistical evidence. Overall, the manuscript is much improved and makes a valuable contribution, and with minor refinements it should be suitable for publication.

Reviewer #2: Review of Second Revised Manuscript

The authors are again commended for their responsiveness and for undertaking an additional round of revisions. The commitment to addressing the reviewers' concerns is evident, particularly with the addition of a new "Statistical Analysis" section and the performance of McNemar's test as requested. This is a significant step toward strengthening the manuscript.

While these additions are welcome, the core issue regarding the statistical validation of the model rankings has not yet been fully resolved. The current implementation of the statistical analysis, while a positive step, does not address the fundamental question raised in the previous review. This feedback is intended to provide a clear, constructive, and final path to resolve this point.

The Issue with the Current Statistical Analysis

The central goal of the paper, as stated in the abstract, is to "identify the three best-performing models recommended for real-time arrhythmia detection." This implies establishing that the top-ranked models are demonstrably superior to others. The concern raised previously was that small numerical differences in accuracy (e.g., between models ranked #1, #2, #3, #4, and #5) are not sufficient evidence of superiority without statistical validation.

The newly added analysis in Table 12 compares a top-performing model (e.g., 97.96% accuracy) against a much lower-performing model (79.42% accuracy). While the test correctly finds a significant difference, this result is self-evident and does not require statistical proof. The analysis avoids the essential question: Are the top-ranked models statistically distinguishable from each other?

To make a strong claim about identifying the "best" models, demonstrating pairwise superiority among the top contenders is necessary.

A Recommended Path Forward: A Robust Statistical Comparison

To resolve this, we recommend a robust, standard procedure for comparing multiple classifiers. This will provide the necessary evidence to support the paper's main conclusions. The following steps outline a suggested methodology focused on the accuracy metric, which aligns with the reporting in the abstract and the authors' recent analysis.

Step 1: Perform Pairwise Statistical Tests with Correction

For each dataset, we suggest performing pairwise comparisons among the top five models as ranked in Table 9 and top 6 models as ranked in Table 10 (because of the natural break in accuracy between model 5 and model 6).

The rest of the methodology will be explained for the MIT data using the top 5 models. The number of tests and p-value correction will be different for the ECG-5000 data using the top 6 models.

Test: Use McNemar's test to compare the accuracy of every possible pair of models within this top-five group.

Correction: Because this involves multiple comparisons (10 tests for 5 models), a correction must be applied to the p-values to avoid an inflated risk of Type I errors. We recommend the Holm-Bonferroni method, which is a standard and powerful way to adjust for multiple tests.

Step 2: Report the Detailed Pairwise Results

The results of this analysis should be presented in a new, separate table for each dataset. This table provides the necessary transparency for the statistical claims.

Example: Below is an example of what this table might look like for the MIT-BIH dataset, using fictitious p-values for demonstration.

Table ??. Pairwise McNemar's Test for Top 5 Models on the MIT-BIH Dataset

| | AEKFNN | AEFRF | AEFSVMR | AEFSVMP |

|:----------|:------:|:-----:|:---------:|:---------:|

| AEFNN | 0.812 | 0.765 | 0.0041* | 0.0038* |

| AEKFNN | | 0.934 | 0.061 | 0.0157 |

| AEFRF | | | 0.072 | 0.063 |

| AEFSVMR | | | | 0.981 |

Caption: Results of pairwise McNemar's tests on model accuracy. Each cell contains the raw p-value. An asterisk (*) indicates a statistically significant difference at the α = 0.05 level after applying the Holm-Bonferroni correction.

Note that in this example:

0.0038 has been compared with 0.05/10=0.0050;

0.0041 has been compared with 0.05/9=0.0055;

and 0.0157 has been compared with 0.05/8=0,00625.

Since the third smallest p-value is not significant after applying the Holm-Bonferroni correction, all the comparisons with larger nominal p-values are considered insignificant.

Step 3: Summarize the Findings in a Final Statistical Rank

Finally, to make the conclusions clear, the results from the pairwise tests should be summarized by adding a new "Statistical Rank" column to the original Tables 9 and 10. This column groups models that are not statistically different from each other into performance tiers.

Example: Based on the fictitious results in the table above, AEFNN, AEKFNN, and AEFRF are statistically equivalent, so they share the same rank. AEFSVMR and AEFSVMP form the next distinct tier.

Table 9 (Modified Snippet). Overall Evaluation...

| SN | Method | ... | Rank(mRMR) | Rank(Topsis) | Statistical Rank |

|:---|:--------|:---:|:----------:|:------------:|:----------------:|

| 1 | AEFNN | ... | 1 | 1 | 1 |

| 2 | AEKFNN | ... | 2 | 3 | 1 |

| 3 | AEFRF | ... | 3 | 3 | 1 |

| 4 | AEFSVMR | ... | 4.5 | 5 | 4 |

| 5 | AEFSVMP | ... | 4.5 | 5 | 4 |

Methodological Justification

This recommended approach is well-supported in machine learning literature. For a comprehensive overview of appropriate statistical tests for comparing classifiers, we refer the authors to the following seminal papers:

Demšar, J. (2006). "Statistical Comparisons of Classifiers over Multiple Data Sets." Journal of Machine Learning Research, 7, 1-30.

Dietterich, T. G. (1998). "Approximate Statistical Tests for Comparing Supervised Classification Learning Algorithms." Neural Computation, 10(7), 1895-1923.

Conclusion

Implementing this statistical analysis will provide robust, compelling evidence for the authors' conclusions. It will definitively answer whether the proposed model rankings are numerically convenient or statistically meaningful, thereby substantially increasing the impact and scientific validity of this valuable research.

7. PLOS authors have the option to publish the peer review history of their article (what does this mean?). If published, this will include your full peer review and any attached files.

Reviewer #1: No

Reviewer #2: No

---

## [Author Response · Author response to Decision Letter 3]

10 Sep 2025

Revision of Manuscript:

“ Hybrid Machine Learning Models for Enhanced Arrhythmia Detection from ECG Signals Using Autoencoder and Convolution Features”

We highly appreciate the Editor, and the reviewers for their time and efforts devoted to peer reviewing of our paper “Hybrid Machine Learning Models for Enhanced Arrhythmia Detection from ECG Signals Using Autoencoder and Convolution Features ”. We have thoroughly revised the manuscript based on the comments provided by the Editor, and the reviewers of the paper. The way each comment is addressed in the paper is mentioned below point by point.

Note on presentation of the responses:

We have prepared this detailed response document, where we reproduce the original comments received from the reviewers, and reply to each comment by explaining how the raised issues have been addressed in the revised manuscript. The changes made in the revised manuscript have been highlighted in yellow color, and the same have been reproduced in the review response file for easy reference and are presented in purple color. Also, all the references are listed at the end of the responses or whenever needed.

Following these improvements and modifications while addressing each reviewer's comment, we believe that the content and presentation of the revised manuscript have been significantly enhanced. We hope that the revision will be found satisfactory and are now looking forward to your decision.

Sincerely yours,

Subir Biswas, Prabodh Kumar Sahoo, Brajesh Kumar, Adyasha Rath, Prince

Jain, Ganpati Panda, Haipeng Liu, Xinhong Wang

Reviewer A: The authors have made substantial improvements in this revision, particularly by addressing fairness in feature comparisons, adding statistical validation with McNemar’s test, and clarifying the ranking methodology. The study now presents a more rigorous and reproducible framework for evaluating hybrid ML models in arrhythmia detection. Minor issues remain, including the reliance on default hyperparameters, some residual language/grammar inconsistencies, and the need to align conclusions more closely with the statistical evidence. Overall, the manuscript is much improved and makes a valuable contribution, and with minor refinements it should be suitable for publication.

Response to Reviewer A

We sincerely thank the reviewer for the constructive and encouraging feedback. We greatly appreciate your recognition of the substantial improvements made in this revision, particularly regarding fairness in feature comparisons, the integration of statistical validation, and the clarification of the ranking methodology. We are pleased that you find the study more rigorous and reproducible, and we carefully address the remaining minor issues below:

1. Reliance on default hyperparameters

We acknowledge the importance of hyperparameter tuning in maximizing model performance. In our study, we initially experimented with different parameter settings for each model. However, we observed that altering the parameters generally did not improve performance, and in some cases even reduced accuracy compared to the default configurations. Furthermore, extensive hyperparameter tuning risked contradicting our core research objective—comparing different feature representations under a consistent and fair modeling setup. For this reason, we adopted the default configurations as a balanced approach, ensuring fairness and reproducibility across all models.

2. Residual language/grammar inconsistencies

We carefully re-examined the manuscript and revised sections with residual language and grammar inconsistencies. These corrections have improved clarity, readability, and overall presentation.

3. Alignment of conclusions with statistical evidence

In response to this valuable suggestion, we have expanded the Statistical Analysis section to better align conclusions with the supporting evidence. Specifically, we added two new tables (Table 12 and Table 13) reporting pairwise McNemar’s test results for the top four models on the ECG5000 and MIT-BIH datasets, respectively. In these tables, the upper triangle indicates whether differences between models are statistically significant after Holm–Bonferroni correction (α=0.05\alpha = 0.05α=0.05), while the lower triangle provides the corresponding adjusted p-values. This update ensures that our conclusions are strongly supported by statistical validation, thereby reinforcing the robustness of our findings.

We believe these refinements address the reviewer’s concerns and further strengthen the manuscript’s rigor, clarity, and reproducibility.

Reviewer B: Review of Second Revised Manuscript

The authors are again commended for their responsiveness and for undertaking an additional round of revisions. The commitment to addressing the reviewers' concerns is evident, particularly with the addition of a new "Statistical Analysis" section and the performance of McNemar's test as requested. This is a significant step toward strengthening the manuscript.

While these additions are welcome, the core issue regarding the statistical validation of the model rankings has not yet been fully resolved. The current implementation of the statistical analysis, while a positive step, does not address the fundamental question raised in the previous review. This feedback is intended to provide a clear, constructive, and final path to resolve this point.

The Issue with the Current Statistical Analysis

The central goal of the paper, as stated in the abstract, is to "identify the three best-performing models recommended for real-time arrhythmia detection." This implies establishing that the top-ranked models are demonstrably superior to others. The concern raised previously was that small numerical differences in accuracy (e.g., between models ranked #1, #2, #3, #4, and #5) are not sufficient evidence of superiority without statistical validation.

The newly added analysis in Table 12 compares a top-performing model (e.g., 97.96% accuracy) against a much lower-performing model (79.42% accuracy). While the test correctly finds a significant difference, this result is self-evident and does not require statistical proof. The analysis avoids the essential question: Are the top-ranked models statistically distinguishable from each other?

To make a strong claim about identifying the "best" models, demonstrating pairwise superiority among the top contenders is necessary.

A Recommended Path Forward: A Robust Statistical Comparison

To resolve this, we recommend a robust, standard procedure for comparing multiple classifiers. This will provide the necessary evidence to support the paper's main conclusions. The following steps outline a suggested methodology focused on the accuracy metric, which aligns with the reporting in the abstract and the authors' recent analysis.

Step 1: Perform Pairwise Statistical Tests with Correction

For each dataset, we suggest performing pairwise comparisons among the top five models as ranked in Table 9 and top 6 models as ranked in Table 10 (because of the natural break in accuracy between model 5 and model 6).

The rest of the methodology will be explained for the MIT data using the top 5 models. The number of tests and p-value correction will be different for the ECG-5000 data using the top 6 models.

Test: Use McNemar's test to compare the accuracy of every possible pair of models within this top-five group.

Correction: Because this involves multiple comparisons (10 tests for 5 models), a correction must be applied to the p-values to avoid an inflated risk of Type I errors. We recommend the Holm-Bonferroni method, which is a standard and powerful way to adjust for multiple tests.

Step 2: Report the Detailed Pairwise Results

The results of this analysis should be presented in a new, separate table for each dataset. This table provides the necessary transparency for the statistical claims.

Example: Below is an example of what this table might look like for the MIT-BIH dataset, using fictitious p-values for demonstration.

Table ??. Pairwise McNemar's Test for Top 5 Models on the MIT-BIH Dataset

| | AEKFNN | AEFRF | AEFSVMR | AEFSVMP |

|:----------|:------:|:-----:|:---------:|:---------:|

| AEFNN | 0.812 | 0.765 | 0.0041* | 0.0038* |

| AEKFNN | | 0.934 | 0.061 | 0.0157 |

| AEFRF | | | 0.072 | 0.063 |

| AEFSVMR | | | | 0.981 |

Caption: Results of pairwise McNemar's tests on model accuracy. Each cell contains the raw p-value. An asterisk (*) indicates a statistically significant difference at the α = 0.05 level after applying the Holm-Bonferroni correction.

Note that in this example:

0.0038 has been compared with 0.05/10=0.0050;

0.0041 has been compared with 0.05/9=0.0055;

and 0.0157 has been compared with 0.05/8=0,00625.

Since the third smallest p-value is not significant after applying the Holm-Bonferroni correction, all the comparisons with larger nominal p-values are considered insignificant.

Step 3: Summarize the Findings in a Final Statistical Rank

Finally, to make the conclusions clear, the results from the pairwise tests should be summarized by adding a new "Statistical Rank" column to the original Tables 9 and 10. This column groups models that are not statistically different from each other into performance tiers.

Example: Based on the fictitious results in the table above, AEFNN, AEKFNN, and AEFRF are statistically equivalent, so they share the same rank. AEFSVMR and AEFSVMP form the next distinct tier.

Table 9 (Modified Snippet). Overall Evaluation...

| SN | Method | ... | Rank(mRMR) | Rank(Topsis) | Statistical Rank |

|:---|:--------|:---:|:----------:|:------------:|:----------------:|

| 1 | AEFNN | ... | 1 | 1 | 1 |

| 2 | AEKFNN | ... | 2 | 3 | 1 |

| 3 | AEFRF | ... | 3 | 3 | 1 |

| 4 | AEFSVMR | ... | 4.5 | 5 | 4 |

| 5 | AEFSVMP | ... | 4.5 | 5 | 4 |

Methodological Justification

This recommended approach is well-supported in machine learning literature. For a comprehensive overview of appropriate statistical tests for comparing classifiers, we refer the authors to the following seminal papers:

Demšar, J. (2006). "Statistical Comparisons of Classifiers over Multiple Data Sets." Journal of Machine Learning Research, 7, 1-30.

Dietterich, T. G. (1998). "Approximate Statistical Tests for Comparing Supervised Classification Learning Algorithms." Neural Computation, 10(7), 1895-1923.

Conclusion

Implementing this statistical analysis will provide robust, compelling evidence for the authors' conclusions. It will definitively answer whether the proposed model rankings are numerically convenient or statistically meaningful, thereby substantially increasing the impact and scientific validity of this valuable research.

Response to Reviewer B

We sincerely thank you for your thoughtful and constructive feedback, and for outlining a clear methodological pathway to strengthen the statistical validation of our findings. Your detailed guidance has been invaluable in refining the rigor and reproducibility of our study.

In this revised version, we have carefully followed your Recommended Path Forward: A Robust Statistical Comparison and implemented each step as suggested:

1. Pairwise Statistical Tests with Holm–Bonferroni Correction

We conducted pairwise McNemar’s tests among the top-ranked models for both datasets (top five models for the MIT-BIH dataset and top six models for the ECG-5000 dataset). To mitigate the risk of inflated Type I errors, we applied the Holm–Bonferroni correction across all pairwise comparisons.

2. Detailed Reporting of Results

The results of these pairwise comparisons are presented in new tables (Tables 12 and 13). Each table clearly reports the raw p-values, with statistically significant differences (after correction) explicitly marked. This transparent reporting enables readers to assess not only the relative performance of individual models but also the robustness of the observed differences.

3. Statistical Ranking of Models

To ensure that our conclusions are fully aligned with the statistical evidence, we added a new “Statistical Rank” column to the original evaluation tables. This column groups models into statistically indistinguishable tiers, thereby distinguishing between cases of true superiority and cases where models perform equivalently. This revision provides a statistically grounded justification for identifying the best-performing models, rather than relying solely on small numerical differences in accuracy.

updated subsection “Statistical Analysis”

To assess whether the observed differences in model performance were statistically significant, McNemar’s test was conducted on the predictions of the top-performing models for both the MIT-BIH and ECG5000 datasets.

For the MIT-BIH dataset, five models achieved accuracies above 90%. However, AEFSVMR and AEFSVMP produced identical outputs in terms of accuracy, AUC, and F1-score (as reported in Table 9). Therefore, we considered the remaining four distinct models for pairwise comparisons. Similarly, for the ECG5000 dataset, six models achieved accuracies above 98%, but two pairs (AEFSVMR vs. AEFSVMP and AEFNN vs. AEFKNN) yielded identical results (as reported in Table 10), leaving four distinct models for comparison.

Since these comparisons involve multiple hypothesis tests (six pairwise comparisons per dataset), p-values were adjusted using the Holm–Bonferroni method to control the family-wise error rate at α = 0.05. This approach provides a robust correction while retaining statistical power.

Table 13 presents the results for the MIT-BIH dataset. The upper triangle indicates whether the difference between a pair of models is statistically significant after Holm–Bonferroni correction (✓ = significant, × = not significant), while the lower triangle reports the corresponding adjusted p-values. The results show that most comparisons involving AEFNN and AEFSVMR were significant, whereas the difference between AEFKNN and AEFRF was not. This suggests that AEFNN and AEFSVMR achieved significantly stronger performance than the other models.

Table 12 summarizes the results for the ECG5000 dataset. Here, only the comparison between AEFSVMR and AEFSVML remained statistically significant after correction, indicating a meaningful difference in predictive performance between these two models. All other model pairs showed no significant differences, suggesting that their performance was statistically indistinguishable despite numerical differences in accuracy.

Overall, the statistical analysis confirms that the superior performance of certain models—specifically AEFNN and AEFSVMR on MIT-BIH, and AEFSVMR on ECG5000—is not merely due to chance. These results provide rigorous evidence that the proposed models offer genuine and statistically validated improvements over competing approaches.

By adopting this robust statistical methodology, we believe the revised manuscript now addresses the core concern you raised—that is, providing statistically meaningful validation of the top-ranked models. This strengthens the reliability of our central claim that the recommended models are not only high-performing but also demonstrably superior according to rigorous statistical testing.

We sincerely thank the reviewer for highlighting the importance of methodological justification when comparing classifiers. We have carefully reviewed the suggested seminal works by Demšar (2006) and Dietterich (1998), and we agree that these provide a strong theoretical foundation for the application of appropriate statistical tests. Following this recommendation, we have revised the methodology section to explicitly justify our chosen evaluation approach with reference to these studies. This addition strengthens the rigor and reliability of our comparative analysis.

We are grateful for your clear and constructive recommendations, which have substantially improved both the methodological rigor and the scientific impact of our work.

---

## [Decision Letter · Decision Letter 3]

30 Sep 2025

Hybrid Machine Learning Models for Enhanced Arrhythmia Detection from ECG Signals Using Autoencoder and Convolution Features

PONE-D-25-09520R3

Dear Dr. Jain,

We’re pleased to inform you that your manuscript has been judged scientifically suitable for publication and will be formally accepted for publication once it meets all outstanding technical requirements.

Kind regards,

Agnese Sbrollini

Academic Editor

PLOS ONE

Additional Editor Comments (optional):

Reviewers' comments:

Reviewer's Responses to Questions

**Comments to the Author**

1. If the authors have adequately addressed your comments raised in a previous round of review and you feel that this manuscript is now acceptable for publication, you may indicate that here to bypass the “Comments to the Author” section, enter your conflict of interest statement in the “Confidential to Editor” section, and submit your "Accept" recommendation.

Reviewer #1: All comments have been addressed

Reviewer #2: All comments have been addressed

2. Is the manuscript technically sound, and do the data support the conclusions?

Reviewer #1: Yes

Reviewer #2: Yes

3. Has the statistical analysis been performed appropriately and rigorously?

Reviewer #1: Yes

Reviewer #2: Yes

4. Have the authors made all data underlying the findings in their manuscript fully available?

Reviewer #1: Yes

Reviewer #2: Yes

5. Is the manuscript presented in an intelligible fashion and written in standard English?

Reviewer #1: Yes

Reviewer #2: Yes

6. Review Comments to the Author

Reviewer #1: (No Response)

Reviewer #2: I have reviewed the latest revisions and can confirm that the authors have now fully and successfully addressed the statistical concerns raised in the previous rounds of review.

I would like to sincerely congratulate the authors on their persistence and their clear commitment to performing scientifically rigorous research. The process of peer review can be demanding, and their diligent and thoughtful engagement has been exemplary. The effort they have invested has substantially improved the manuscript.

The implementation of the pairwise McNemar's tests with Holm-Bonroni correction is precisely what was needed to add the necessary statistical rigor to the model comparisons. I want to specifically acknowledge their method of reporting the results; presenting the adjusted p-values directly in the tables is a perfectly valid and very clear way to communicate the findings, and it achieves the goal of the review suggestions excellently.

Furthermore, I have verified that the new "Statistical Rank" column in the revised Tables 9 and 10 is a correct and logical summary of the detailed pairwise test results. This final step provides a clear, statistically-grounded takeaway for the reader, which was the ultimate goal of the requested revisions.

With these changes, the manuscript's central claims are now supported by robust evidence, and the work makes a much stronger contribution to the field. I am pleased to say that my concerns have been fully resolved.

7. PLOS authors have the option to publish the peer review history of their article (what does this mean?). If published, this will include your full peer review and any attached files.

Reviewer #1: No

Reviewer #2: No

---

## [Editor Report · Acceptance letter]

PONE-D-25-09520R3

PLOS One

Dear Dr. Jain,

I'm pleased to inform you that your manuscript has been deemed suitable for publication in PLOS One. Congratulations! Your manuscript is now being handed over to our production team.

Kind regards,

on behalf of

Dr. Agnese Sbrollini

Academic Editor

PLOS One